# Distilled Circuits: A Mechanistic Study of Internal Restructuring in Knowledge Distillation

**Reilly Haskins**                                                    *reilly.haskins@pg.canterbury.ac.nz*
*Department of Computer Science and Software Engineering*
*University of Canterbury*

**Benjamin Adams**                                                    *benjamin.adams@canterbury.ac.nz*
*Department of Computer Science and Software Engineering*
*University of Canterbury*

**Reviewed on OpenReview:** *https://openreview.net/forum?id=S1KJE2ZW64*

## Abstract

Knowledge distillation compresses a larger neural model (teacher) into smaller, faster student models by training them to match teacher outputs. However, the internal computational transformations that occur during this process remain poorly understood. We apply techniques from mechanistic interpretability to analyze how internal circuits, representations, and activation patterns differ between teachers and students. Focusing on GPT2 and its distilled counterpart DistilGPT2, and generalizing our findings to both bidirectional architectures and larger model pairs, we find that student models can reorganize, compress, and discard teacher components, often resulting in a stronger reliance on fewer individual components. To quantify functional alignment beyond output similarity, we introduce an alignment metric based on influence-weighted component similarity, validated across multiple tasks. Our findings reveal that while knowledge distillation preserves broad functional behaviors, it also causes significant shifts in internal computation, with important implications for the robustness and generalization capacity of distilled models.

## 1 Introduction

Knowledge distillation (KD) compresses neural models by training a smaller student model to replicate the outputs of a larger teacher, enabling comparable performance with fewer parameters and more efficient deployment in terms of computation, memory, and inference time (Hinton et al., 2015; Sanh et al., 2020). Despite its widespread use, the internal transformations that occur during the KD process remain poorly understood (Stanton et al., 2021; Baek & Tegmark, 2025; Zhang et al., 2023).

While prior work has focused on optimizing knowledge transfer between teacher and student, less attention has been given to the internal mechanisms that emerge in the student. Distilled models often develop alternative computational strategies to approximate teacher behavior with fewer parameters (Wu et al., 2024). These mechanisms may improve efficiency but can also undermine generalization by relying on heuristics that diverge from those of the teacher. A mechanistic understanding of these transformations is essential for evaluating the robustness and functional alignment of student models.

We apply techniques from mechanistic interpretability (MI) to analyze how knowledge distillation restructures internal circuits, representations, and activation patterns, and propose a metric for use in automatically quantifying the functional alignment between teacher and student. Our analysis centers on GPT2 (Radford et al., 2019) as the teacher and DistilGPT2 (HuggingFace, 2019b) as the student. Here, the student (6 layers, 12 heads, 82M parameters) has around two-thirds of the parameters of the teacher (12 layers, 12 heads, 124M parameters). We chose a relatively smaller model pair to focus our study on because larger models introduce complications when it comes to extracting and comparing full circuits, a granularity that we wished to look

at in this work. In addition to our primary analysis on the GPT2 and DistilGPT2 pair, we replicate our methodology on BERT (Devlin et al., 2019) and DistilBERT (Sanh et al., 2020) as well as Llama-3.1-8B (Dubey et al., 2024) (Llama) and Llama-3.1-Minitron-4B-Depth-Base (Sreenivas et al., 2024) (Minitron) to assess generalization of our methodology and findings to both architectural and model size changes. These secondary studies reveal consistent restructuring patterns and changes in robustness, suggesting that the internal transformations produced by distillation reflect broader trends not limited to a single model family. Through this investigation, we ask the following questions:

- How do the internal representations and circuits that emerge in the student differ from those in the teacher during KD?

- What effect does KD have on the robustness of the internal mechanisms and components in the student model?

- How accurately can an alignment metric quantify functional alignment in internal computation between a teacher and student model?

By providing a mechanistic understanding of the changes that take place during KD, this research aims to enhance confidence in the use of distilled models. Understanding how internal computation is restructured during the KD process will help in designing reliable student models and mitigating potential failure cases. Full code implementation for all experiments is visible in the supplied repository [1].

## 2 Background and related work

### 2.1 Knowledge distillation

Knowledge distillation (KD) compresses neural models by transferring knowledge from a larger teacher model to a smaller student model. This is typically done by softening the teacher and student logits via a temperature-scaled softmax (Hinton et al., 2015):

$$p_i^T(x) = \frac{\exp(f_i(x)/T)}{\sum_{j=1}^K \exp(f_j(x)/T)} \tag{1}$$

to produce probability distributions over the output classes. These softened distributions are then used in the distillation loss, which minimizes the Kullback-Leibler (KL) divergence between the teacher and student outputs over a dataset $\mathcal{D}_{\text{distill}}$:

$$\mathcal{L}_{\text{KD}} = \mathbf{E}_{x \sim \mathcal{D}_{\text{distill}}} \left[ D_{\text{KL}} \left( p_t^T(x) \, \| \, p_s^T(x) \right) \right] \tag{2}$$

Higher values of $T$ reveal finer-grained similarity between logits.

Recent work has investigated what is learned during KD, revealing both behavioral and representational gaps between teachers and students. Wu et al. (2024) show that standard distillation often promotes simplicity bias, leading students to rely on spurious correlations rather than faithfully replicating teacher reasoning mechanisms. Techniques like Jacobian matching (Srinivas & Fleuret, 2018) and contrastive representation distillation (Tian et al., 2020) help reduce this bias but still do not guarantee full mechanism transfer. While Wu et al. (2024) evaluate knowledge transfer through behavioral counterfactual evaluations, a deeper mechanistic understanding of internal model computations remains open.

Several factors such as model capacity and output structure have been shown to play a critical role in shaping distillation outcomes (Cho & Hariharan, 2019; Phuong & Lampert, 2019; Stanton et al., 2021). Cho & Hariharan (2019) show that overly complex teachers may hinder student learning due to representational bottlenecks, and propose early stopping to produce "softer" outputs. Optimization challenges further

---

[1] https://github.com/Reih02/distilled_circuits

complicate distillation. Stanton et al. (2021) find that even with sufficient capacity, students may struggle to match teacher distributions due to hard-to-optimize objectives. This suggests that distillation often acts more as implicit regularization than exact imitation. Complementing these empirical findings, theoretical work by Phuong & Lampert (2019) identifies class separability, optimization bias, and strong monotonicity as key factors explaining why soft labels aid distillation.

From a geometric view, Baek & Tegmark (2025) use sparse crosscoders (Lindsey, 2024) to demonstrate that KD reshapes large language model (LLM) feature spaces, leading to specialized reasoning mechanisms. This finding aligns closely with our mechanistic focus by demonstrating that a student may reproduce teacher predictions while relying on different underlying mechanisms, potentially undermining robustness and generalization. Understanding these internal shifts is critical for assessing the out-of-distribution behavior of distilled models and for improving evaluation techniques, such as performance difference analysis (Sanh et al., 2020; Jiao et al., 2020), neural model search algorithms (Trivedi et al., 2023), and unified performance benchmarks (Yang et al., 2024).

## 2.2 Mechanistic interpretability

Mechanistic interpretability (MI) aims to reverse-engineer neural networks by identifying how they compute and represent information (Bereska & Gavves, 2024; Olah et al., 2020). In transformer models, attention heads often specialize in modeling syntactic or semantic relationships between tokens, whereas the multi-layer perceptrons (MLPs) tend to capture higher-level, abstract features, often acting as implicit detectors of specific concepts or patterns (Templeton et al., 2024; Bereska & Gavves, 2024). These components interact in structured circuits, analyzable via causal interventions, ablation studies, and activation analysis (Conmy et al., 2023; Meng et al., 2022).

MI has successfully uncovered complex functional circuits within language models. For instance, Wang et al. (2023) identified a 26-head circuit in GPT2 for the indirect object identification (IOI) task, decomposing its functional roles. To address the scalability challenge of discovering such circuits, Conmy et al. (2023) introduced Automatic Circuit DisCovery (ACDC), a pruning-based method that automatically recovers important subgraphs by measuring the impact of edge ablations on output logits.

MI has also been applied to understand how circuits evolve over time. Wang et al. (2025) show that fine-tuning alters edge connectivity while preserving node identity, proposing a circuit-aware low-rank adaptation (LoRA) method to improve the fine-tuning process. Notably, their focus is structural, emphasizing edge changes rather than functional shifts in circuit roles. Similarly, Nanda et al. (2023) study circuit emergence during grokking, a phenomenon where models initially memorize before abruptly generalizing (Power et al., 2022). Proposing metrics to track the stabilization of mechanistic structure over training, they demonstrate that internal computation evolves through measurable stages.

Causal tracing (Meng et al., 2022) and causal scrubbing (Chan et al., 2022) offer hypothesis-driven validation of circuit functions within a single model but require manual design and lack scalability across tasks and architectures. In contrast, our proposed alignment metric (Section 4) provides an automated, influence-weighted comparison of functional components between models, capturing task-relevant differences without relying on hand-crafted hypotheses. This makes it more practical for cross-model analyses such as those within KD.

Collectively, these studies show that KD reshapes both outputs and internal dynamics, and that MI can reveal these changes. Our work addresses the gap in mechanistic understanding of the KD process by applying mechanistic tools to the domain of KD, tracing computational transformations and representation changes during distillation.

## 2.3 Tasks

Below, we describe the tasks studied in this paper, all of which were introduced in prior work. Performance of each model on each tested task is reported in Appendix B.

### 2.3.1 Sequence completion

**Numeral sequence completion**   Lan et al. (2024) introduce a task involving a sequence of four monotonically increasing numeral elements (e.g., 1, 2, 3, 4), with each numeral prepended with non-sequence tokens ("Van done in 1. Hat done in 2 ..."). This is done to encourage the identification of a circuit representation that can also achieve the subtask of selecting sequence members from non-sequence members. Each example ends with non-sequence members, such that the next token to be predicted will be the final sequence element. Thus, the model must output a single token to complete the sequence.

**Word sequence completion**   Similarly, Lan et al. (2024) produce a variant of the numeral sequence completion task that uses number words in place of digits (e.g., "four, five, six, seven" instead of "4, 5, 6, 7"). This serves as a robustness check that components implementing certain roles, for example the successor operation, generalize and are not reliant on digit-specific tokenization artifacts or memorized numeral strings.

### 2.3.2 Indirect object identification

Wang et al. (2023) produce a dataset alongside their analysis of the indirect object identification (IOI) problem, in which the model is tasked with identifying the indirect object in a sentence. An example here is the sentence "When Mary and John went to the store, John gave a bottle of milk to ", where the following correct token would be "Mary" due to the given context.

### 2.3.3 Question answering

For the question answering task, we make use of the SimpleQA dataset (Wei et al., 2024). This dataset involves short, fact-seeking questions and corresponding answers sourced from around the internet. Categories include science and technology, geography, sports, history, and more. An example from this dataset is the query "Who received the IEEE Frank Rosenblatt Award in 2010?", with the answer being "Michio Sugeno".

## 3   Case study: Numeral sequence completion

In this section, we perform a case study analyzing the degree of transfer of internal mechanisms, including circuits and representations, between a teacher (GPT2, 124M parameters) and student model (DistilGPT2, 82M parameters) on the numeral sequence completion task (Lan et al., 2024). To test whether our findings extend across architectures, we also study BERT and DistilBERT (referred to collectively as BERT models), as well as Llama and Minitron (Llama models) on the same task, with further details in Appendices I and H, respectively. See Appendix A for details of the training process used to produce these student models. Across model families, we observe consistent trends, where students place greater reliance on fewer components, compress multiple functions into single heads or MLPs, and omit less critical functionalities. While some architecture-specific differences exist (e.g., no first-token divergence in the BERT models' MLPs), the core restructuring behaviors remain consistent. This suggests that the transformations induced by KD are not generally model-specific. To further support this, we include a complementary case study on the IOI task in Appendix G, where similar patterns of transfer are again observed on the GPT model pair.

### 3.1   Methodology

The numeral sequence completion task requires predicting the next token given a sequence $x_1, \ldots, x_t$, i.e., learning the conditional probability distribution $P(x_{t+1} \mid x_1, \ldots, x_t)$. Following Lan et al. (2024) and Conmy et al. (2023), we identify circuits via iterative pruning and path patching, using a corrupted dataset (Meng et al., 2022) to isolate each component's influence on the task. After identifying the circuit components for the task, we determine their roles through query-key attention analysis for heads and representation analysis for MLPs, enabling a detailed comparison of how teacher and student networks perform the task. Experiments here used 100 examples and were run on a single NVIDIA A100 GPU in under one hour, with data obtained from Lan et al. (2024).

**Compute/scaling.** Let $N$ be the number of examples in our dataset, $C$ the number of components (heads and MLPs), and $E$ the number of candidate edges tested. Component (node) ablations scale $\mathcal{O}(C \cdot N)$, while edge/path-patching can scale up to $\mathcal{O}(E \cdot N)$, where E is $\mathcal{O}(C^2)$ in the dense fully-connected case.

### 3.1.1 Circuit discovery

For both node and edge discovery, we follow prior work and quantify performance shifts using the *logit difference*, defined as the difference between the pre-softmax scores (logits) of the correct token and a designated distractor (incorrect) token. In our experiments, the distractor is the final element of the input sequence:

$$\Delta\ell = \ell_{correct} - \ell_{incorrect} = log(\frac{p_{correct}}{p_{incorrect}}) \tag{3}$$

where $p_{token}$ denotes the model's predicted probability for the token. A larger $\Delta\ell$ indicates a stronger preference for the correct token, reflecting high confidence in the task. A negative value instead indicates that the model prefers the incorrect token to the correct one. It is worthwhile to note here that, due to the logits being pre-softmax scores, a logit difference of $x$ implies the correct token is $e^x$ times as likely as the incorrect one.

Circuit nodes (defined as MLPs and attention heads) are ablated by replacing their activations with corrupted means, preventing downstream use. Iteratively ablating MLPs and attention heads layer-by-layer (backward then forward), we retain components whose ablation causes a performance drop exceeding a threshold $T_n = 0.20$, i.e., a drop of at least 20% of the original logit difference, forming the final set $C$ of important nodes (see Appendix N for a sweep of threshold values and impacts on circuit size and interpretability metrics to address robustness of our subsequent findings). To capture critical interactions between nodes, we extend activation patching with path-patching: ablating outgoing edges to isolate those that significantly contribute to task performance (where performance drops below $T_n$), following approaches similar to Hanna et al. (2023) and Lan et al. (2024).

**Interpretability metrics: completeness, faithfulness, minimality.** Beyond recovering a performant subgraph, this procedure enforces standard interpretability essentials by design, by retaining only components whose ablation causes a $\geq 20\%$ drop in logit difference for the task. On the numeral sequence task, retaining only the extracted circuit and ablating the rest of the GPT2 / DistilGPT2 models reduces mean logit difference by just 18.99% (teacher) and 19.88% (student), indicating completeness with respect to our chosen threshold of 20.00%.

Conversely, ablating the circuit while leaving the remainder intact causes a catastrophic drop in logit difference, from 6.12 to -0.28 in the teacher and from 3.75 to -0.31 in the student, demonstrating faithfulness (i.e., the circuit is necessary for the model's behavior on this task).

Finally, minimality follows from the pruning rule, where every retained component individually passes the $\geq 20\%$ ablation threshold; removing any one element therefore drops the circuit below the faithfulness criterion.

### 3.1.2 Component comparison

We manually compare the functionality of in-circuit components (attention heads and MLPs) across models. For attention heads, we compute and inspect the query–key (QK) matrices, which reveal token-level self-attention patterns and information flow throughout the task (Vaswani et al., 2017). After identifying the role of each head, we match and contrast key heads between the teacher and student models based on functional similarity.

For MLPs, we apply residual-stream decomposition to attribute changes in logit difference to the MLP at each token (Appendix F). To identify teacher–student matches, we summarize each layer's activations with PCA (Appendix D, Figure 7): from the centered activation matrix (tokens × features), we compute the SVD of its covariance and retain the top three principal components. This captures the dominant directions of variance

while suppressing irrelevant cross-model noise. We then compute the mean absolute cosine similarity between corresponding components. The resulting score is a compact, noise-robust proxy for functional overlap, where values near 1 indicate closely aligned dominant computations and values near 0 indicate orthogonal roles.

### 3.1.3 Role validation

Beyond attention-pattern and representation analyses, we validate component roles using two complementary techniques that test causality and linear accessibility:

**Activation patching (causal test):** We construct corrupted prompts by randomizing the numeral sequence while preserving the token distribution, then pass both corrupted and clean prompts through the model. We patch clean activations back into the corrupted run at a fine granularity (component × token), and measure logit-difference recovery for the correct next token. This isolates which components and tokens are causally sufficient for the behavior. We use both whole-block patching and path-specific patching (e.g., QK vs. OV) to localize effects to particular attention head subcircuits, with all recoveries being normalized within each layer.

**Linear probing (representational test):** We fit simple, regularized linear probes on the residual stream at each layer to predict task-relevant variables (e.g., the $i$-th numeral or next numeral) from position-specific activations. Probes are trained with held-out validation, balanced class sampling, and permutation controls to guard against frequency and leakage artifacts. A sharp increase in probe accuracy at a given layer indicates that the information has become linearly decodable there, consistent with proximal upstream components writing that signal into the residual stream.

Together, activation patching and probing provide more quantitative and causal evidence about which components perform certain tasks and where it becomes accessible to downstream computation to confirm inferred component roles. We provide further head-specific results and probe training details in Appendix J.

### 3.2 Findings

We successfully identify circuits for the numeral sequence completion task across both the teacher and student networks. We wish to note here that the below specific component roles and restructuring effects observed serve as an exemplified mechanistic decomposition for this specific task and model pair, which may not generalize exactly to other tasks or models. However, we do observe similar (but not exact) effects in a different, natural-language based task (Appendix G) as well as on different model pairs (Appendices I and H). Full circuit diagrams are provided in Appendix E.

#### 3.2.1 Attention analysis

Analyzing query-key attention matrices reveals significant changes in how the student model performs sequence completion. We organize our analysis by key attention head functionalities. Note that an attention head from layer $x$, head $y$ is denoted as L$x$-H$y$, prefixed by 'T-' (teacher) or 'S-' (student). We follow Lan et al. (2024) and report ablation-induced performance change percentage, defined as the logit difference in the ablated model relative to the unablated model:

$$\Delta P\% = 100 \cdot \frac{\Delta \ell_{ablated} - \Delta \ell_{base}}{|\Delta \ell_{base}|} \tag{4}$$

Where a negative value corresponds with performance worse than baseline, e.g. a performance change of -50.00% corresponds with half of the original logit difference.

**Similar member detection (T-L1-H5).** We identify a head in the first layer of the teacher (T-L1-H5) that detects repeated elements by attending to past mentions (Appendix C, Figure 4). No corresponding behavior is observed in the student head, suggesting a loss of this inductive bias during distillation. One possible explanation for this discrepancy is that the component does not play a crucial role across many tasks

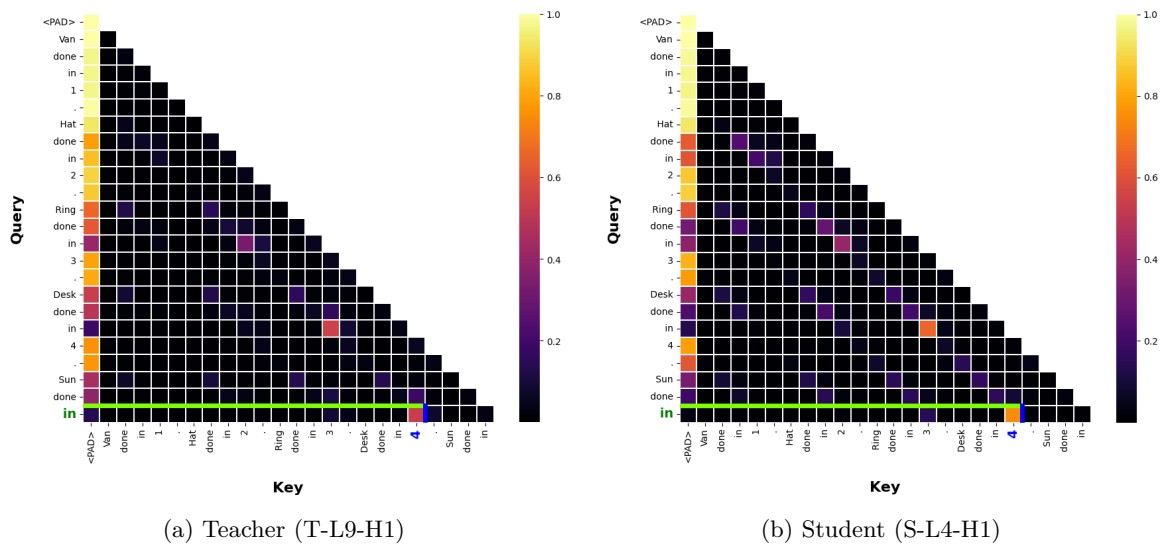

(a) Teacher (T-L9-H1)  (b) Student (S-L4-H1)

Figure 1: Teacher (left) and student (right) QK attention matrices for the successor head functionality

and was therefore removed as part of the trade-off between functionality and parameter efficiency. Similar behavior was observed between the BERT models (Appendix I), further suggesting that this functionality is not crucial and is favoured to be omitted by student models. It is worth noting that, although often described as "similar member detection" in prior work, our probes indicate it primarily encodes the previous numeral, and similar members are a secondary function (see Appendix Section J.4.1).

**Numeral detection (T-L4-H4 / S-L2-H4).**  Both teacher (T-L4-H4) and student (S-L2-H4) were seen to detect and encode the numeral sequence via high self-attention between numerals and their predecessors (Appendix C, Figure 5). However, the student relies more heavily on a single head for this behavior (performance change: -87.73% vs. teacher's -33.18%), indicating compressed reliance due to parameter constraints and potential for less robust behavior in the case of distributional shifts.

**Numeral mover (T-L7-H11 / S-L3-H11).**  Numeral mover heads were found across both models, which transfer numeral information to the final token (Appendix C, Figure 6). The student was seen to replicate this behavior accurately, ensuring numerals are encoded into the head's output and propagated via the residual stream. However, the student again shows significantly higher reliance (performance change: -72.83% vs. teacher's -41.64%). This finding reinforces the fact that the student lacks backup heads (Wang et al., 2023), relying more heavily on single components due to its reduced parameters. Further decomposition reveals both networks use these heads primarily for their OV-circuits, which are responsible for determining what information to move to the residual stream (Nanda et al., 2023).

**Successor computation (T-L9-H1 / S-L4-H1).**  Heads attending from the final token to the last numeral in the teacher are preserved well in the student (Figure 1), but with a stronger self-attention score between the final token and the numeral in the student (0.71 vs. 0.55) and again greater reliance (performance change: -77.57% vs. teacher's -34.94%). This pattern of increased attention and reliance further supports our hypothesis that the student model is less robust due to its reduced parameter capacity.

### 3.2.2 MLP analysis

Layer-wise attribution, where differences in adjacent residual streams between layers are computed, reveals only a few MLPs significantly impact performance on this task across both models (Appendix F). Cosine similarities and PCA are used to identify MLP pairs with shared functionality, as well as some teacher MLPs with no apparent counterpart in the student. Note that we refer to the $x$-th MLP in the teacher as 'MLP-T-$x$', and likewise we use 'MLP-S-$x$' when referring to the student.

**MLP-T-8.** This MLP component was seen to steer the teacher's output toward the last given element rather than the correct next one (increasing the logit difference by 0.146 when ablated). This behavior is absent in the student, as indicated by cosine similarities with MLP-S-3 and MLP-S-4 being low (0.240 and 0.360, respectively). This omission of counterproductive behavior, which improves performance in this case, is consistent with previous findings that KD can act as an implicit regularizer by filtering out noisy or suboptimal behaviors learned by the teacher. (Jooste et al., 2022).

**MLP-T-9 / MLP-T-10 / MLP-S-4.** MLP-S-4 merges the functionality of MLP-T-9 and -10 (cosine similarity: 0.634 and 0.687). PCA and unembedding of the residual stream (logit lens) reveal clear structural similarities and emergence of correct predictions at layer 9/10 in the teacher and layer 4 in the student (Figure 2a). This evidence confirms that both MLP-T-9 / MLP-T-10 and MLP-S-4 are responsible for computing the next numeral in the sequence. These MLPs share a layer with corresponding successor heads (T-L9-H1 / S-L4-H1), indicating they likely use the final numeral and increment it to generate the next. Together, these findings suggest that the student has effectively merged the functionality of MLP-T-9 and MLP-T-10 into MLP-S-4, learning a more efficient representation in terms of parameter count. This observed compression supports the hypothesis that KD can compress redundant computations while preserving functionality (Jooste et al., 2022; Wu et al., 2024).

**MLP-T-11 / MLP-S-5.** These components show a representational divergence for the first token, with a cosine similarity of -0.302, while other tokens remain similar (mean cosine similarity 0.721). This late-stage divergence suggests the student optimizes its representation under parameter constraints, potentially impacting performance on tasks where the initial token contributes disproportionately to the output (Ferrando et al., 2022; Vaswani et al., 2017). PCA captures this difference effectively, as the first token's embeddings are positioned at opposite corners in PCA space (Figure 2b), while other tokens cluster closely.

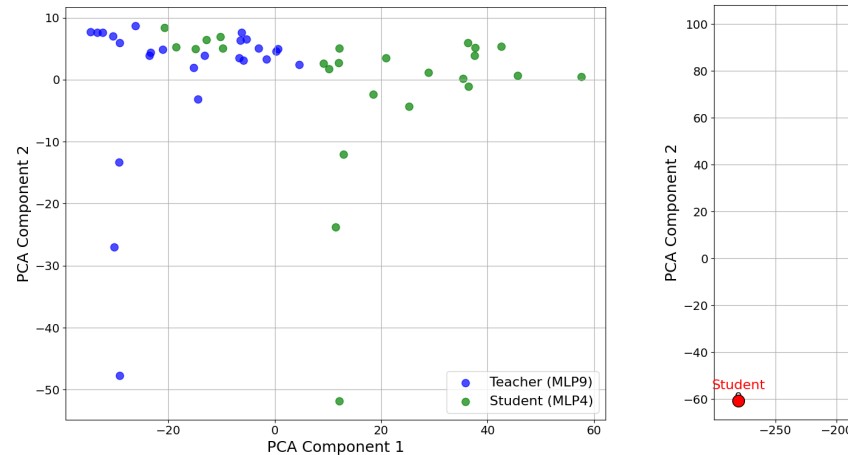 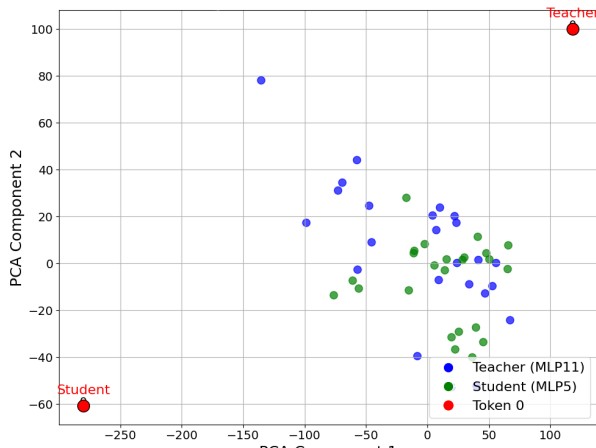

(a) PCA projection of MLP activations within MLP-T-9 and MLP-S-4, showing structural similarity

(b) PCA projection of MLP activations within MLP-T-11 and MLP-S-5, with highlighted first token

Figure 2: Comparison of PCA projections across different teacher-student MLP pairs.

### 3.2.3 Robustness to component ablation

Table 1 shows that the student model is more vulnerable to key attention head ablations (where the activations are corrupted as per the methodology outlined in Subsection 3.1), due to its higher reliance on a smaller set of critical components compared to the teacher. This trend holds beyond this circuit and model pair, where we see that across other pairs, the student consistently sees a significantly larger mean drop in performance under component ablation (See Table 2 and Appendix K). While the teacher distributes functionality across multiple heads, the student's brittle reliance aligns with observations by Hendrycks et al. (2020) that distilled models, although often matching in-distribution accuracy, often exhibit significantly reduced robustness

Table 1: Comparison of performance changes caused by ablating highly-influential attention heads across teacher and student models.

| Functionality | Teacher | Student |
|---|---|---|
| Similar member detection head | $-27.83\%$ | $-$ |
| Numeral detection heads | $-33.18\%$ | $-87.73\%$ |
| Numeral mover heads | $-41.64\%$ | $-72.83\%$ |
| Successor heads | $-34.94\%$ | $-77.57\%$ |

Table 2: Mean ablation-induced drops in performance across model pairs with 95% bootstrap CIs on the numeral sequence completion task across 384 random examples. Larger values = less robustness to ablation

| Pair | Teacher | Student | Difference | Alignment Score |
|---|---|---|---|---|
| Llama | 0.8395 [0.5943, 1.1553] | 2.2014 [1.5841, 2.9286] | 1.3619 | 0.9778 |
| GPT | 3.0561 [1.7097, 4.6601] | 12.2401 [6.4496, 18.8281] | 9.1840 | 0.9452 |
| BERT | 6.2644 [3.4185, 9.5738] | 16.8856 [10.7452, 23.5468] | 10.6212 | 0.8872 |

and increased vulnerability to distribution shifts and input corruptions. This fragility arises from the lower parameter count, which limits availability of fallback mechanisms.

## 4    Alignment metric

In the previous section, we have shown that the KD process can produce students with significant internal differences to their teacher and decreased robustness to component ablation. However, this style of analysis is extremely time-consuming and intractable for larger models and/or more complex circuitry. Motivated by this, we now focus on automating the comparison of functional alignment (i.e., similarity in the internal behaviors that support task performance) by proposing an alignment metric that quantifies the extent to which functionally important components in both models behave similarly in their contribution to the task. This enables a scalable evaluation of how KD restructures computation, helping assess trade-offs between parameter efficiency and the preservation of functional behavior.

### 4.1    Methodology

The metric is computed in three steps. First, we assign influence scores to each component (attention heads and MLPs). Next, we match components between teacher and student via representational similarity. Lastly, alignment is computed by summing similarity-weighted influence agreement across matched pairs, weighted by similarity. This effectively penalizes functional divergence while tolerating unmatched, low-impact components.

**Scalability.**   Influence computation requires one ablation per component, i.e. $\mathcal{O}(C \cdot N)$ patched forward passes. Matching requires $\mathcal{O}(C \cdot N)$ work to compute dataset-aggregated activation summaries (e.g., mean activations), followed by $\mathcal{O}(C^2)$ pairwise cosine-similarity comparisons (for $C_T \times C_S$ teacher-student pairs).

#### 4.1.1    Calculating component influence

For each component, $c \in \{\text{attention head}, \text{MLP}\}$ within each model, we calculate an *influence score*, reflecting the component's contribution to task performance. To determine this, we measure the change in the model's logit difference that results from ablating the component, using the path patching technique described earlier. We then normalize these scores by dividing each component's change in logit difference by the largest change observed across the model, so that all influence scores fall between 0 and 1. In the (rarely observed) case of a component *improving* the logit difference under ablation, we clamp these negative values to 0. Note that this not only serves as a measure of component importance, but also in measuring the similarity of robustness

distributions between models, an important measure if the goal of KD is to produce reliable students. Formally, let $m \in \{T, S\}$ denote a model, $D = \{x^{(n)}\}_{n=1}^N$ the dataset, and $\bar{\Delta}\ell(m) := \frac{1}{N}\sum_{n=1}^N \Delta\ell_m(x^{(n)})$ the model's average logit difference on $D$. For a component $c \in C_m$ (attention head or MLP), let $m \setminus c$ denote the model with $c$ ablated via path patching. Then the (normalized) influence score is

$$I_m(c) = \frac{\max\left(0, \ \bar{\Delta}\ell(m) - \bar{\Delta}\ell(m \setminus c)\right)}{\max_{c' \in C_m} \max\left(0, \ \bar{\Delta}\ell(m) - \bar{\Delta}\ell(m \setminus c')\right)}. \tag{5}$$

### 4.1.2 Matching components

To align teacher and student components, we define a component similarity score and match each teacher component to its nearest neighbor in the student (within the same component type), allowing many-to-one matches. We compute similarity $S$ from dataset-mean activations for attention heads, and from the leading eigenvectors of the activation covariance for MLPs (computed via SVD).

Formally, for a component $c \in C_m$, let $a_{m,c}(x) \in \mathbb{R}^{L \times d}$ denote its token-wise matrix of $d$-dimensional activations on input example $x$, and define the dataset-mean activation $\bar{a}_{m,c} := \frac{1}{N}\sum_{n=1}^N a_{m,c}(x^{(n)})$. For MLPs, let $u_{m,c}[k]$ be the $k$-th leading eigenvector from the covariance matrix of the centered activations of $c$ (over all tokens and all $x^{(n)} \in D$). We define the similarity between a teacher component $c_T$ and student component $c_S$ by

$$S(c_T, c_S) = \begin{cases} \cos\left(\text{vec}(\bar{a}_{T,c_T}), \ \text{vec}(\bar{a}_{S,c_S})\right), & c_T \in C_T^{\text{head}}, \ c_S \in C_S^{\text{head}}, \\ \frac{1}{3}\sum_{k=1}^3 \left|\cos\left(u_{T,c_T}[k], \ u_{S,c_S}[k]\right)\right|, & c_T \in C_T^{\text{mlp}}, \ c_S \in C_S^{\text{mlp}}. \end{cases} \tag{6}$$

The teacher-to-student matching is then

$$\pi(c_T) = \arg\max_{c_S \in C_S} S(c_T, c_S), \qquad M = \{(c_T, \pi(c_T)) \mid c_T \in C_T\}, \tag{7}$$

where matching is performed within each component type.

### 4.1.3 Calculating model alignment

Once we have obtained influence scores and similarity values for matched components, we compute an overall alignment score, $A$, to quantify how closely the student replicates the teacher's internal functional behavior on a task. $A$ is defined as the similarity-weighted average influence agreement across all matched components $M$, emphasizing functional alignment where both representational similarity and task-specific influence agree. Components with low similarity contribute less to the numerator but still count in the denominator, meaning that poorly aligned pairs reduce the overall score. This focuses the metric on functional consistency in shared mechanisms, while implicitly discouraging representational mismatches. Formally:

$$A_{T,S} = \frac{1}{|M|} \sum_{(c_T, c_S) \in M} S(c_T, c_S) \cdot (1 - |I_T(c_T) - I_S(c_S)|). \tag{8}$$

Where $S_i$ is the similarity between student and teacher component activations, and $|I_T(c_T) - I_S(c_S)|$ is the absolute difference between their normalized influence scores. Subtracting the influence difference from one transforms it into a similarity measure, rewarding closer functional alignment with a higher score. Because influence scores are normalized, $A$ captures *relative* computational alignment and remains robust across differences in model size and architecture. Unlike alternative model comparison methods such as CKA (Kornblith et al., 2019) or SVCCA (Raghu et al., 2017), which measure global representational similarity, our metric accounts for task-specific functional importance. This allows it to detect changes in critical circuits that those methods may miss, providing a more targeted measure of mechanistic alignment.

This metric is grounded in the principle that a student trained to replicate a teacher's capabilities should internalize and reproduce the teacher's internal computations (Aguilar et al., 2020). Misalignments in influence

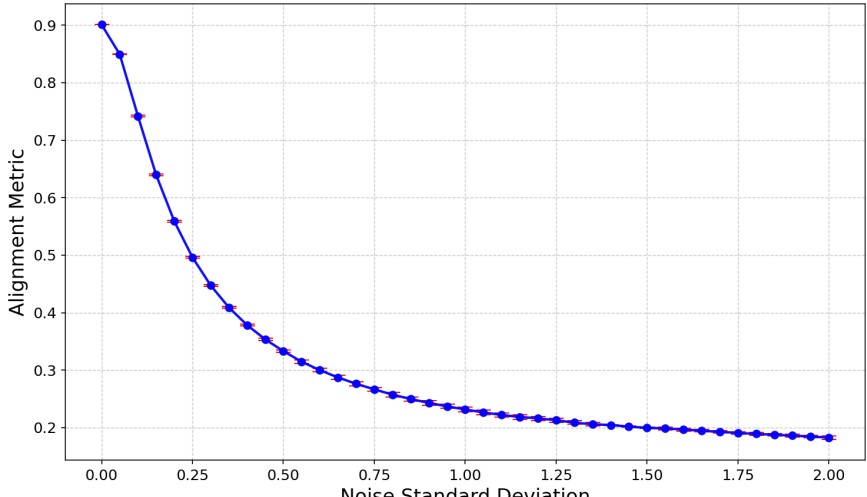

Figure 3: Results of our activation noise-injection experiment on our proposed alignment metric for the numeral sequence completion task on GPT2, with error bars shown in red.

among functionally similar components indicate deviations in behavior, potentially impacting generalizability and reliability. By rewarding functional agreement, the alignment score effectively captures the degree of task-specific circuit replication between models. See Appendix M for sensitivity analyses of the key design choices underlying the alignment metric.

## 4.2 Findings

To validate our metric and hypotheses, we conduct two experiments:

- Ablation experiment: We simulate a series of student models with varying degrees of functional alignment to the teacher by introducing Gaussian noise to the student's activations during numeral sequence completion, verifying that our alignment score decreases as functional divergence increases.

- Cross-task comparison: We compare teacher-student alignment scores across a variety of tasks and assess their correlation with performance gaps. This tests whether functional alignment tracks real-world task performance.

### 4.2.1 Ablation experiment

To validate the alignment metric, we run an ablation experiment by injecting Gaussian noise into the student model's activations on the numeral sequence completion task (data obtained from Lan et al. (2024)), simulating increasing functional misalignment. Noise is drawn from a zero-mean distribution with standard deviation from 0.0 to 2.0 in increments of 0.05. Each noise sample is evaluated across five random seeds, and we report error bars indicating one standard deviation across these seeds.

We observe an inverse-logarithmic relationship between our metric and the standard deviation of injected noise (Figure 3). This is expected, as greater noise leads to lower alignment, confirming the metric's sensitivity to misalignment and its utility in comparing student models. The plateau in alignment score around 0.2 suggests that beyond a certain noise level, the student behaves randomly, and further perturbations have little effect. Remaining alignment likely stems from residual structure or token-level priors.

### 4.2.2 Cross-task comparison

We evaluate alignment across four tasks using GPT2 and DistilGPT2 as the primary teacher–student pair, and include both BERT / DistilBERT and Llama / Minitron (see Appendix H) results on select tasks to assess

Table 3: Alignment and logit-difference metrics across tasks. Parentheses show 95% CIs.

| Task | Model pair | Alignment | $\Delta\ell$ | Mean $\ell$ |
|------|-----------|-----------|--------------|-------------|
| Question answering | Llama / Minitron | 0.9812 (0.9790, 0.9832) | 0.7157 (0.1324, 1.2991) | 2.8848 (2.5931, 3.1765) |
| Numeral sequence completion | Llama / Minitron | 0.9778 (0.9761, 0.9784) | 0.6943 (0.6188, 0.7674) | 4.5233 (4.4345, 4.6116) |
| Numeral sequence completion | GPT2 / DistilGPT2 | 0.9452 (0.9445, 0.9457) | 2.2019 (2.0702, 2.3341) | 5.0220 (4.9523, 5.0895) |
| Word sequence completion | GPT2 / DistilGPT2 | 0.9404 (0.9355, 0.9422) | 5.5281 (5.4153, 5.6476) | 1.0869 (1.0169, 1.1546) |
| Indirect object identification | GPT2 / DistilGPT2 | 0.9182 (0.9163, 0.9186) | 3.6974 (3.5593, 3.8389) | 1.3129 (1.2301, 1.3958) |
| Numeral sequence completion | BERT / DistilBERT | 0.8872 (0.8814, 0.8917) | 0.8824 (0.8229, 0.9415) | 0.2761 (0.2227, 0.3301) |
| Indirect object identification | BERT / DistilBERT | 0.8803 (0.8761, 0.8807) | 1.8022 (1.5674, 2.0214) | 2.2498 (2.1357, 2.3639) |
| Word sequence completion | BERT / DistilBERT | 0.8413 (0.8412, 0.8415) | 0.6555 (0.5623, 0.7540) | -0.4068 (-0.4560, -0.3602) |
| Numeral sequence completion | GPT2 / DistilBERT | 0.7654 (0.7629, 0.7677) | 6.2880 (6.2079, 6.3683) | 2.9789 (2.9344, 3.0207) |

Mean $\ell$ is the average of the mean teacher and student logit differences across the dataset.

architectural generalization. Full results are shown in Table 3. For numeral and word sequence completion, we use data from Lan et al. (2024) containing both digit- and word-based sequences, with 384 examples. For indirect object identification (IOI), we follow Wang et al. (2023) and use 500 randomly-sampled examples requiring the identification of indirect objects in sentences. For question answering, we use SimpleQA (Wei et al., 2024) across 200 random samples (see Appendix H.2 for more detailed findings).

These results demonstrate that performance gaps ($\Delta\ell$) are not reliable indicators of functional alignment between models by themselves. For example, on the numeral sequence completion task between GPT2 and DistilGPT2, we observed a high alignment score of 0.95, while the $\Delta\ell$ was 2.20 (with both models being able to solve the task confidently, GPT2: 6.12, DistilGPT2 3.92). On the same task between BERT and DistilBERT, we see a lower $\Delta\ell$ of 0.88 (BERT: 0.71, DistilBERT: -0.17), despite the fact that the alignment score is significantly lower at 0.89. In this case, we attribute this lowered alignment score in the BERT pair due to the greater levels of noise and less distinct specialization in student components (see Appendix I for more details). This reveals that alignment is more tightly linked to internal computation patterns and representations than surface-level behavioral performance, and thus commonly used methods for assessing similarity between teacher and student via performance differences alone across some dataset (Sanh et al., 2020; Jiao et al., 2020; Yang et al., 2024) may yield misleading conclusions for OOD cases. Similarly, on the word sequence completion task, the BERT pair score much lower in alignment than the GPT2 pair (0.84 vs. 0.94), despite performance being closer within the BERT pair ($\Delta\ell = 0.66$ vs. 5.53). This provides further evidence that the output behavior can conceal significant internal differences in computation.

**Alignment between a mismatched pair.** Alignment between GPT2 and DistilBERT is substantially lower on the numeral sequence completion task (0.7654), consistent with their architectural differences (autoregressive vs. bidirectional). These structural mismatches, in addition to the fact that they do not form a teacher-student pair (and hence DistilBERT was not optimized to produce GPT2's logit distribution), naturally lead to differing implementations of functionality, as discussed in Appendix I.

**Alignment between a larger pair.** On the Llama (8B parameters) / Minitron (4B parameters) model pair, we observe the highest alignment scores of 0.9778 (numeral sequence completion, AppendixH.1) and 0.9812 (SimpleQA, AppendixH.2), which correspond with some of the lowest $\Delta\ell$ across all pairs and tasks, suggesting more similar internal computations. It is also worth noting here that this model pair shows the smallest difference in mean ablation-induced drops in performance on the numeral sequence completion task (Table 2), which naturally increases their alignment score due to the $(1 - |I_T(c_T) - I_S(c_S)|)$ term.

One likely explanation for this is that, for such a simple task, increasing parameters above 4B may not lead to a benefit in terms of capacity for implementation of needed functionality. Looking at studying distilled students of larger parameter counts across more diverse and complex tasks is a valuable direction for future work here. Across these examples, the metric's sensitivity to alignment loss across a variety of architectures and its capability to reflect expected similarities and differences in computation, e.g. in the case of the diverging architecture of GPT2 / DistilBERT, supports its use across model families. Note that the procedure is model-agnostic, but the resulting alignment score is task-conditioned because influence is defined with respect to a task.

### 4.3 Using the alignment metric in practice

Given a teacher $T$ and one or more candidate students $S_j$ of potentially differing parameter size / architecture / training style, one may wish to know which student's internal task-specific computational pathways best approximate the teacher. As demonstrated in Section 4.2, comparison of task accuracy between models alone is not always an effective predictor of similar internal computation. For this reason, we recommend computing alignment between the teacher and each candidate student model on the intended deployment-relevant task(s), and using this as an additional selection signal. We expect this to be particularly useful when accuracy between candidates is similar, as the alignment metric in this case would be primarily measuring the degree of unfaithful computational shortcuts induced in the student.

In practice, this process yields a per-task alignment score $A(T, S_j)$ that is comparable across model sizes due to influence normalization. When selecting among candidate students with comparable task performance, prefer higher $A$ to prioritize preservation of task-relevant internal computation. We also recommend reporting $A$ alongside a robustness summary, such as mean component ablation-induced performance drop (Table 2), since students often rely more heavily on a smaller set of components.

Because the matching step requires pairwise comparisons between components ($\mathcal{O}(C^2)$), applying the alignment metric to larger models can become cumbersome. In practice, it may be useful to restrict matching to the top-$K$ most influential components, reducing the matching cost to $\mathcal{O}(K^2)$, while still providing a practical first-pass estimate.

## 5 Conclusion

We applied mechanistic interpretability methods to study how internal computation changes during knowledge distillation (KD) across different tasks, focusing on GPT2 and DistilGPT2 and extending findings to both bidirectional architectures (BERT) and larger models (Llama) to show generalizability of findings. While student models were seen to generally preserve broad functional behavior from their teacher, they can significantly restructure internal circuits by compressing, reorganizing, and discarding components. This often leads to increased reliance on fewer critical components, reducing robustness to ablation and distributional shifts.

To quantify these changes, we introduced an alignment metric that captures functional similarity beyond output behavior, validated through controlled experiments. Our findings show that performance similarity alone is not a reliable indicator for similarity of internal computation, and that circuit-level analysis is essential for understanding functional alignment in distilled models. This metric can support safer use of compressed models in low-resource settings, and our findings also highlight the risk of brittle internal mechanisms causing failures in high-stakes applications without rigorous evaluation. The specific circuit decompositions and internal restructuring deep dives should be read as task- and model-specific case studies, whereas the compression/reliance/omission trends are the aspects we expect to transfer more broadly.

## 5.1 Future work and limitations

Our findings open several avenues for future research while also highlighting current limitations. We selected models strategically to maximize inter-model differences and thereby improve generalizability of our findings, but our study remains constrained by the relatively small set of models evaluated (6). A natural extension is to apply our methodology and alignment metric to a broader, more diverse collection of models that vary in architecture, training regime (e.g. dataset), and parameter scale. In particular, clarifying the relationships between alignment and robustness and parameter count on a wider range of complex tasks and across differing distillation datasets may reveal more information about whether there exist reliable scaling laws here. We believe this would be useful to inform practitioners and researchers when choosing or configuring student models for particular domains.

Automation of the component role attribution procedure discussed in Section 3.1.2 would also be a valuable direction to investigate, as it would allow for more practical reproduction of our model comparison methodology on a broader set of models and analyses. We believe the use of interpretability agents (Schwettmann et al., 2023; Shaham et al., 2024) would be valuable for this direction.

Additionally, although our alignment metric was shown to capture broad functional similarity effectively, increasing its granularity and interpretability would improve its usefulness for selecting and tuning distilled models to meet specific deployment requirements. Future work could also explore ways to incorporate model diffing techniques such as the Sparse Crosscoder (Lindsey, 2024), which has shown promise for producing shared sets of features across models. Exploring ways to incorporate the metric during the distillation process (e.g., as a loss term to discourage the student learning computational shortcuts) may also be fruitful.

Lastly, our findings motivate future work developing theoretical accounts of when and why knowledge distillation preserves, merges, or re-routes internal computations, and how these capacity-driven changes relate to redundancy and robustness. This direction could yield clearer conditions and predictions for when the phenomena we observe should persist (or fail) in more complex and realistic settings.

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

# A  Student model training details

Our experiments analyze teacher-student pairs using publicly released student checkpoints. Because the datasets and loss terms used during distillation training can influence which computational mechanisms are preserved or discarded, we document below the details of each student model used in this work.

## A.1  DistilGPT2

We use the publicly released DistilGPT2 checkpoint (HuggingFace, 2019b) as our primary student model. The model card reports training on the same corpus as GPT-2, OpenWebTextCorpus (Gokaslan & Cohen, 2019), using the GPT-2 tokenizer, with a standard knowledge distillation process as described in Section 2.1, alongside the standard LM loss

## A.2  DistilBERT

For our BERT replication study, we use a publicly released DistilBERT checkpoint (HuggingFace, 2019a). The DistilBERT report indicates distillation on the same corpus as BERT (English Wikipedia (wik) and

Table 4: Baseline performance (logit difference) of each studied model across tasks (N=100).

| Model | Numeral Seq. | Word Seq. | IOI | Question Answering |
|---|---|---|---|---|
| GPT2 | 6.1127 | 3.8077 | 3.2645 | - |
| DistilGPT2 | 3.7530 | -1.6371 | -0.2888 | - |
| BERT | 1.1650 | -0.1295 | 5.2709 | - |
| DistilBERT | 0.4904 | -1.3064 | 3.6949 | - |
| Llama-3.1-8B | 3.5375 | - | - | 3.4218 |
| Llama-3.1-Minitron-4B | 3.9062 | - | - | 2.6686 |

Toronto BookCorpus (Zhu et al., 2015)) with dynamic masking and without the next-sentence prediction objective, alongside a standard distillation-based training loss.

### A.3 Llama-3.1-Minitron-4B-Depth-Base

For our Llama replication study, we use Llama-3.1-Minitron-4B-Depth-Base (Sreenivas et al., 2024). The model card describes this model as produced by pruning Llama-3.1-8B (Dubey et al., 2024) (depth pruning) followed by continued training with standard logit-based distillation over 94B tokens using the Nemotron-4 15B continuous pretraining data corpus (Parmar et al., 2024).

## B  Model performance

In Table 4, we report observed performance across all model, task pairs studied in our work, across 100 randomly-sampled examples from their respective datasets. Performance is reported as the logit difference between the correct and incorrect tokens, consistent with all other reportings of performance in this paper.

## C  Attention head visualizations

### C.1  QK attention matrix for T-L1-H5 (similar member detector)

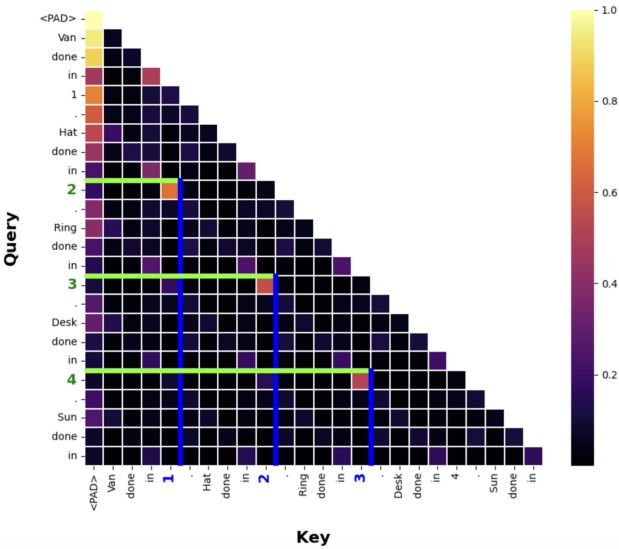

Figure 4: Query-key attention matrix for T-L1-H5 (similar member detection)

The QK attention matrix for T-L1-H5 can be seen in Figure 4. Note the high self-attention activations between each numeral and the previous numeral in the sequence, as well as duplicates of words such as 'in'.

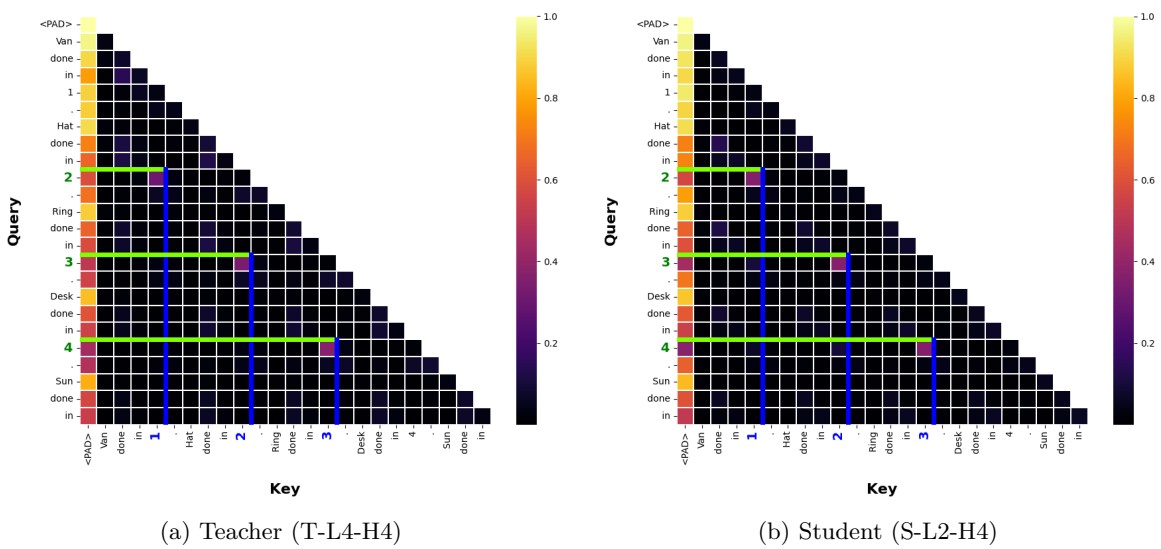

(a) Teacher (T-L4-H4)  (b) Student (S-L2-H4)

Figure 5: Teacher (left) and student (right) QK attention matrices for the numeral detection functionality

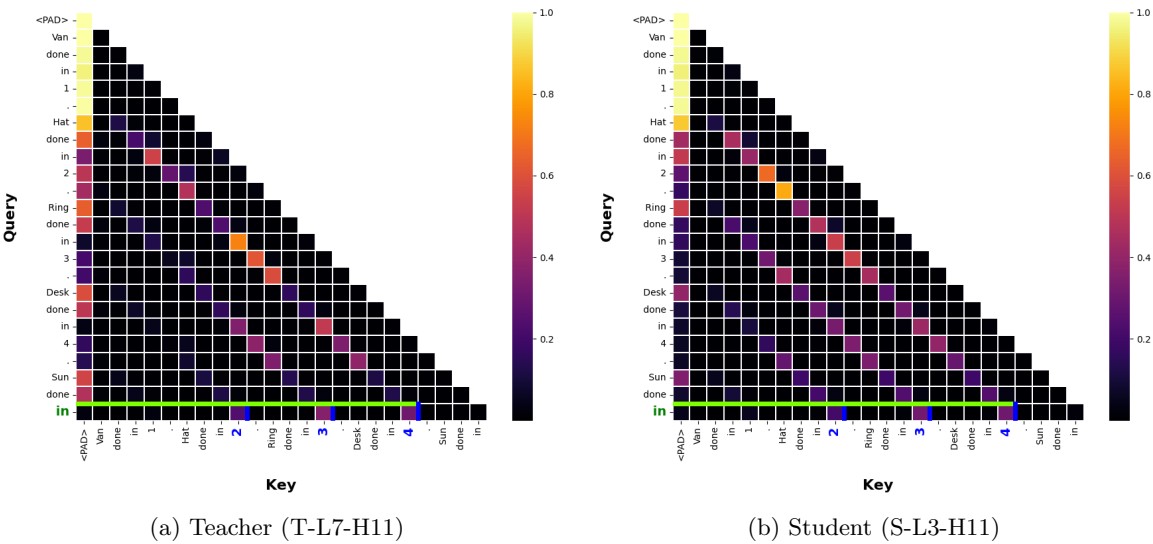

(a) Teacher (T-L7-H11)  (b) Student (S-L3-H11)

Figure 6: Teacher (left) and student (right) QK attention matrices for the numeral mover functionality

## C.2   QK attention matrices for T-L4-H4 / S-L2-H4 (numeral detectors)

Numeral detectors were found across both models, which are responsible for tracking numerals (Figure 5). This manifests as a high self-attention between each numeral in the sequence and the previous numeral.

## C.3   QK attention matrices for T-L7-H11 / S-L3-H11 (numeral movers)

The numeral mover functionality can be seen in more detail in Figure 6. These heads are responsible for aggregating information about the numerals of the sequence into the final token, for use in downstream tasks.

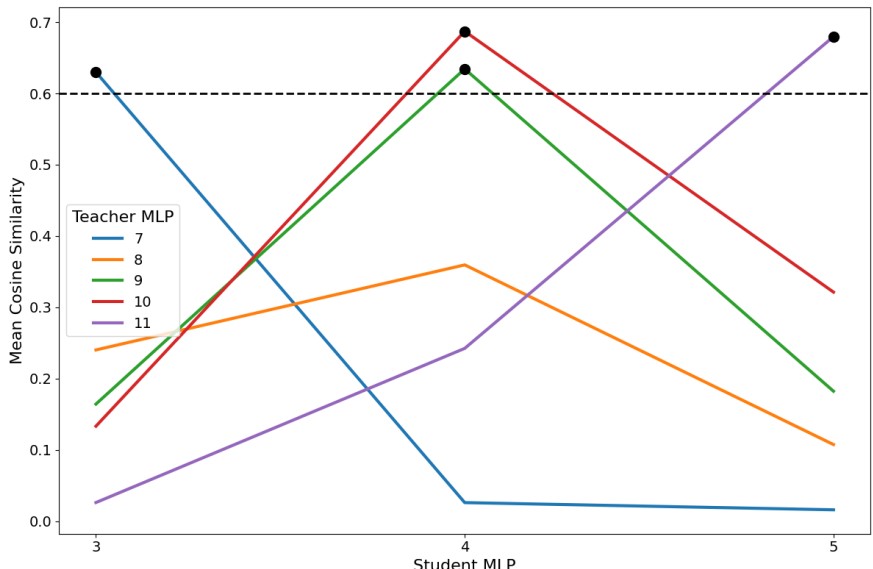

Figure 7: Mean cosine similarities of MLP activations between teacher and student networks, with pair similarities above 0.6 highlighted

## D  MLP comparisons

### D.1  Mean cosine similarity between MLPs

Figure 7 shows the cosine similarities between various components of the teacher and student MLPs. The plot reveals multiple high-similarity pairs, particularly in the mid-to-late layers of the two networks. Additionally, the degree of similarity appears to correlate with the depth of the MLP layers across networks, suggesting similar levels of computation abstraction progression.

## E  Numeral sequence completion circuits

Figures 8 and 9 show the full identified circuits (including head decompositions into query (q), key (k), and value (v) components) for the GPT2 and DistilGPT2 teacher-student pair on the numeral sequence completion task. Notably, the student's circuit is much smaller, with significantly fewer components than the teacher's. These figures were obtained using code supplied by Lan et al. (2024)

## F  MLP layer-wise residual stream analysis

Here, we present Figures 10 and 11, which illustrate how we determined token-level importance of MLP components across teacher and student models. Notably, MLP-0 exhibits a significant influence on the residual stream in both models, primarily due to its role in transforming token embeddings before attention mechanisms are applied. However, we exclude MLP-0 from further analysis in this case, as its contribution is largely tied to input representation rather than task-specific computation, which is the focus of our investigation. In the figures, MLPs 9 and 10 in the teacher and MLP 4 in the student show high residual stream contribution on the final token, reflecting importance for the task.

## G  Complementary case study: Indirect object identification

Here we present our methodology and findings of our complementary case study on the indirect object identification (IOI) task in more detail. We follow the same methodology as defined in the numeral sequence

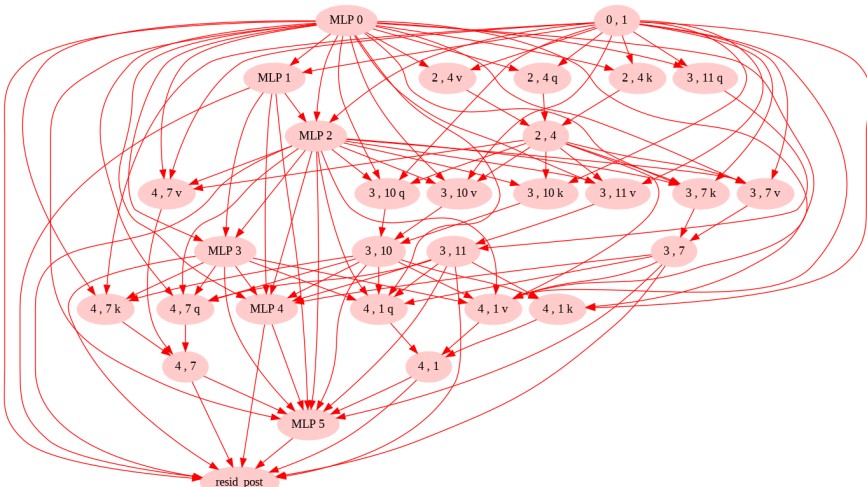

Figure 8: Complete identified circuit for the teacher network. Attention heads are denoted as either query, key, or value components.

Figure 9: Complete identified circuit for the student network. Attention heads are denoted as either query, key, or value components.

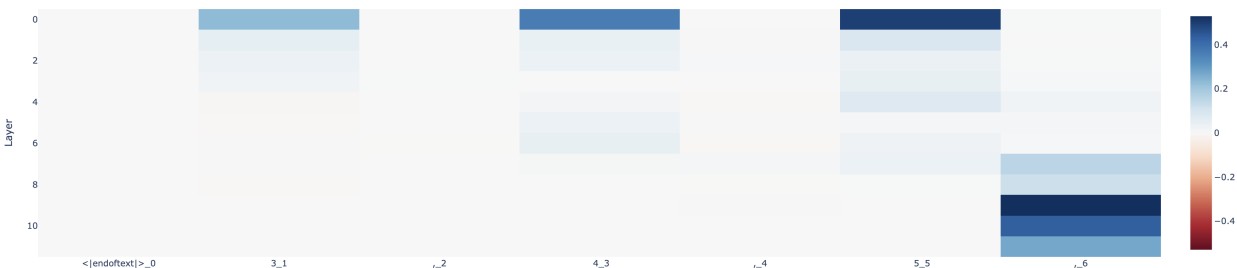

Figure 10: Teacher's layer-wise residual stream contribution per MLP

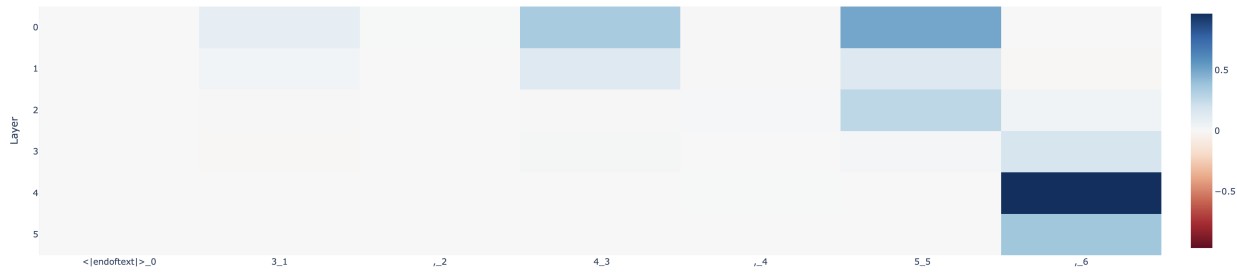

Figure 11: Student's layer-wise residual stream contribution per MLP

completion task case study (Section 3), making use of GPT2 as the teacher and DistilGPT2 as the student. Due to the low utilization of MLPs by either model for this task, we focus our study on attention head components. We identify various attention heads present in the student model which appear to copy across functionality effectively from the teacher model (originally identified by Wang et al. (2023)), although with similar differences to those seen in the numeral sequence completion case study. We outline each functionality in more detail below.

For the purposes of this case study, we choose a critical subset of those functionalities identified by Wang et al. (2023), and choose a single representative head from the teacher for each task, chosen by taking the head responsible for the largest contribution to its functionality.

**Performance differences.**    We measure a large discrepancy in IOI task performance between the teacher and student, with the teacher achieving a logit difference of 3.26 and the student achieving -0.29, suggesting that the teacher can perform the task reliably, while the student cannot.

**Duplicate token heads (T-L0-H1 / S-L0-H5).**    This head serves to pay strong self-attention between each token and itself. Note that the student has roughly copied this across, but shows greater self-attention between duplicate tokens than the teacher does, along with more noise in the activations (Figure 12). This finding is consistent with those in the numeral sequence completion case study, showing evidence that the student often relies more heavily on fewer parameters for the task.

**Name mover heads (T-L9-H9 / S-L5-H2).**    We find loose matches for the name mover head in the student, a crucial functionality identified by Wang et al. for the IOI task. This head is responsible for copying across the indirect object name to the final token, resulting in prediction of the correct answer. The QK matrices (Figure 13) show that the activation patterns are structurally similar in the student for this head, but the student has significantly lower attention values on the indirect object, suggesting less-confident predictions. This discrepancy in the functionality of a core head is likely a source of the lower performance on this task from the student when compared to the teacher.

**Previous token heads (T-L4-H11 / S-L2-H11).**    The previous token head is responsible for paying high self-attention from each token to the previous token. Wang et al. suggest that this head helps the model

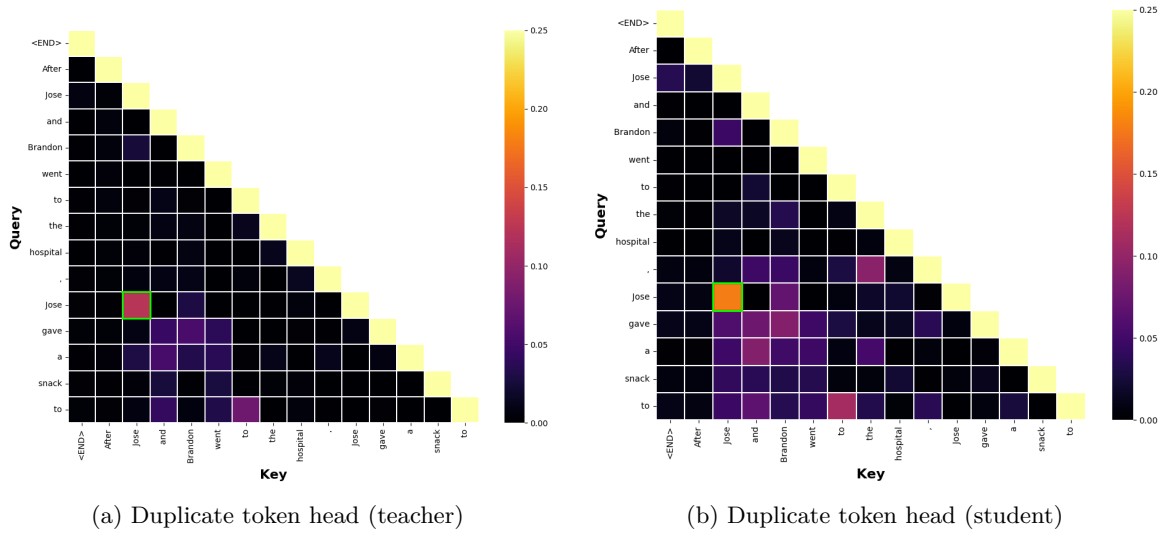

(a) Duplicate token head (teacher)    (b) Duplicate token head (student)

Figure 12: Duplicate token head QK matrices across teacher (left) and student (right)

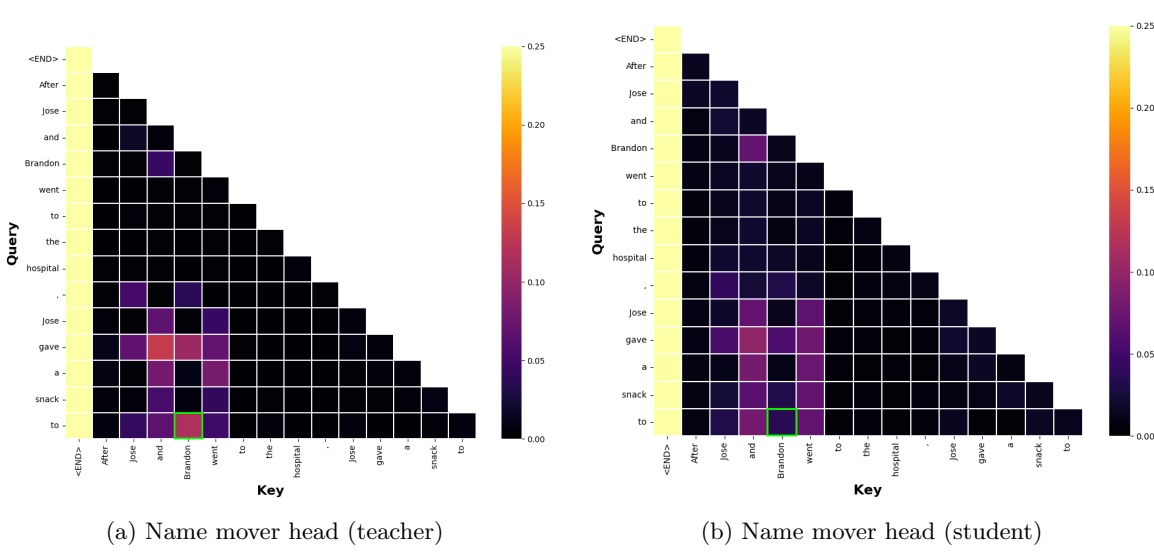

(a) Name mover head (teacher)    (b) Name mover head (student)

Figure 13: Name mover head QK matrices across teacher (left) and student (right)

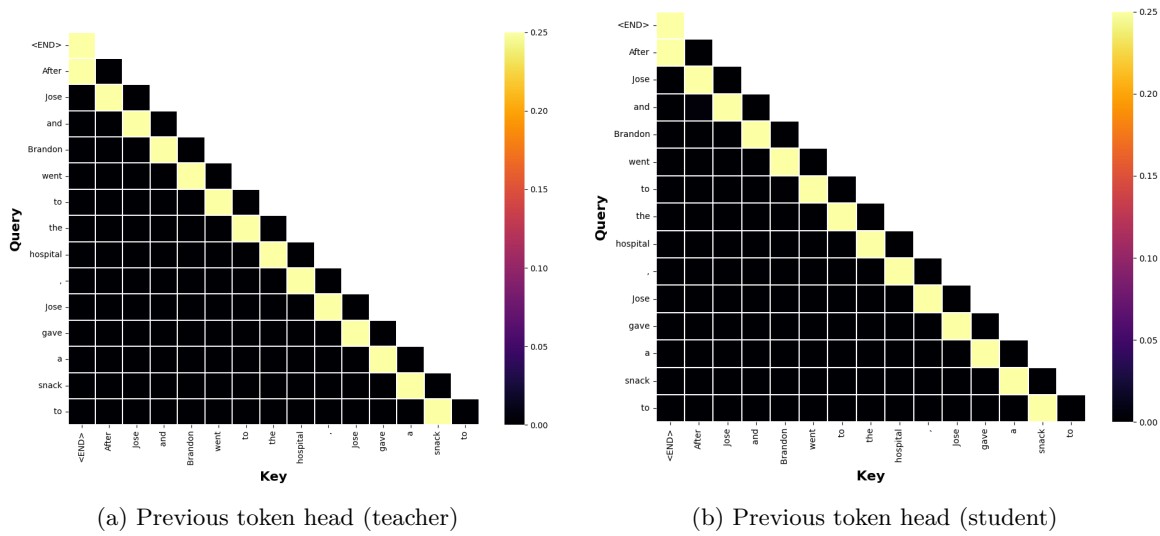

(a) Previous token head (teacher)      (b) Previous token head (student)

Figure 14: Previous token head QK matrices across teacher (left) and student (right)

to maintain and propagate model information through the network early in processing. We find a strong presence of this functionality in the student model, shown through almost identical activation structure and values (Figure 14).

**Induction heads (T-L5-H5 / S-L3-H10).** The induction head was seen to attend to the earlier occurrence of the name token if it occurs twice, creating a shortcut path allowing downstream heads (e.g name mover head) to access information from the first appearance of the correct name. For our analysis of this head in the student, we find a similar pattern to that seen in the name mover head, with the student copying across the broad structure of the induction head from the teacher, but with significantly smaller activation values. This suggests that the student has again failed to copy across the functionality with high confidence, likely degrading performance in the IOI task.

**Ablation performance drop per component.** Here, we show the performance drop (as a percentage of the unablated performance) caused by ablating different attention heads across the two models for the IOI task in Figure 15. Notably, the student contains numerous components which cause performance to drop significantly when ablated, while most components do not cause a notable performance drop when ablated in the teacher. This supports our findings from the numeral sequence completion task (Section 3) and BERT / DistilBERT study (Appendix I), where we saw that the student generally puts significantly higher reliance on individual components than the teacher.

## H Llama-3.1-8B and Llama-3.1-Minitron-4B-Depth-Base replication study

### H.1 Numeral sequence completion

To assess the external validity of our robustness and alignment findings on more modern, higher-parameter transformers, we partially replicate the numeral-sequence case study on larger models. A full case-study replication is out of scope here because identifying and comparing the complete circuit becomes substantially more complex with the increased number of components and layers. We study Llama-3.1-8B (teacher; (Dubey et al., 2024)) and Llama-3.1-Minitron-4B-Depth-Base (student; (Sreenivas et al., 2024)). Minitron is a depth-pruned, distilled derivative of Llama-3.1-8B, where the number of layers is reduced from 32 to 16 by selecting an optimal subset that minimizes accuracy loss on the WinoGrande benchmark (Sakaguchi et al., 2021). Hidden dimensionality and the number of attention heads per layer are held constant with the teacher, as in the other model pairs in this study.

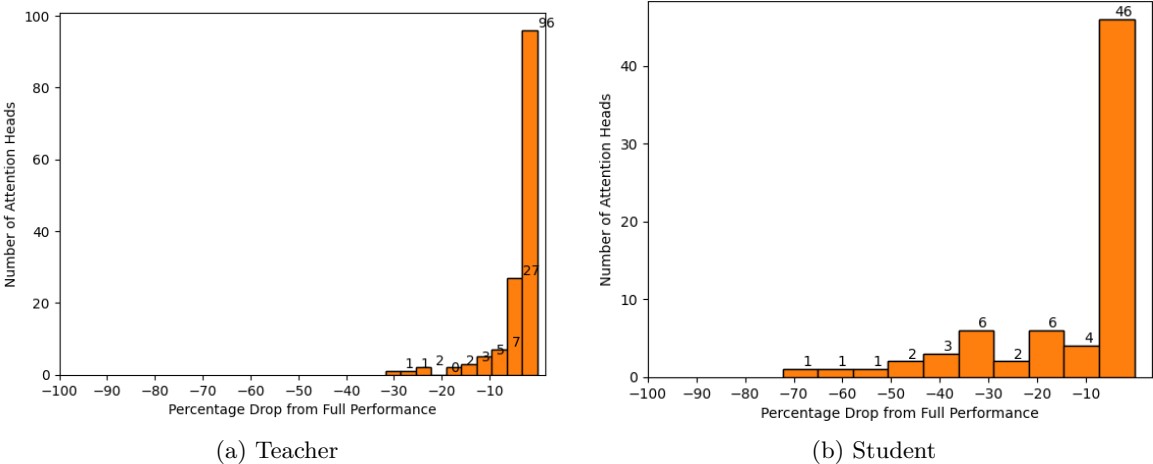

(a) Teacher                (b) Student

Figure 15: Distribution of performance drops caused by ablation across attention heads in both the teacher (left) and student (right) BERT models

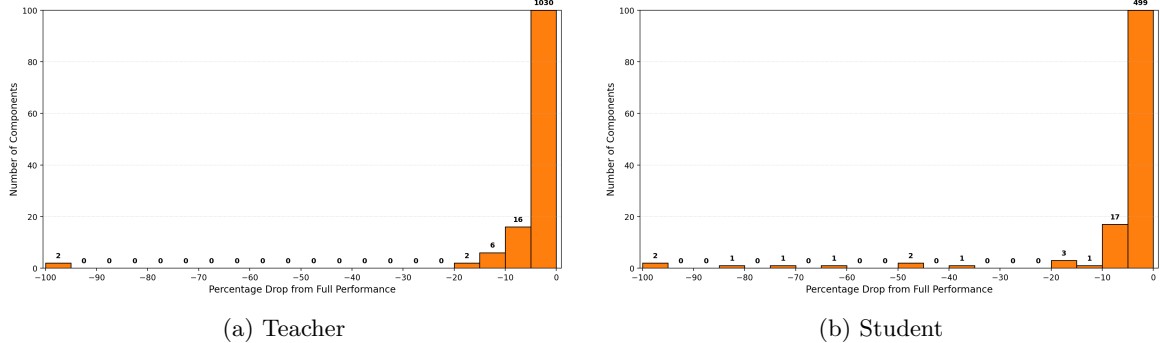

(a) Teacher                (b) Student

Figure 16: Distribution of performance drops caused by ablation across components in both the teacher (left) and student (right) Llama models

Interestingly, we observe a slightly higher logit difference in the student (4.87) than in the teacher (4.18). In all other pairs we examined, the teacher exhibited the larger logit difference. These values are similar, which may reflect convergence of higher-parameter models on a simple task such as numeral-sequence completion.

Within this parameter range, we again find strong evidence that distilled student models are less robust than their teachers under component ablation. The student exhibits larger performance drops from individual component ablations than the teacher (Figure 16). Across all components, the mean ablation-induced performance drop in the teacher is 0.84 [0.59, 1.15], while in the student it is 2.20 [1.58, 2.93] (95% CIs), where the non-overlapping CIs indicate a clear between-model difference. Additionally, eight components in the student cause a drop of at least 20%, compared with just two in the teacher. The two components that produce total (100%) performance collapse in both models are the MLPs at layers 0 and 1, suggesting these early-layer MLPs are critical for the task (consistent with early feature formation). Other student components producing declines > 20% are MLPs 13 (−35.30%), 14 (−47.06%), and 15 (−80.61%), together with attention heads L14H1 (−45.64%), L13H27 (−64.48%), and L15H22 (−74.35%). No other components in the teacher cause a drop exceeding 20%.

Although the student's reduced ablation robustness is statistically significant, the effect size is smaller than in other pairs (Appendix K). The student's average ablation drop exceeds the teacher's by 9.18 percentage points in the GPT2 pair and by 10.62 in the BERT pair, but by only 1.36 in the Llama pair. This contrast is consistent with the higher alignment score for the Llama pair (0.98) relative to GPT2 (0.95) and BERT (0.89): by construction, our alignment metric increases with the similarity of influence distributions. We view

testing the relationship between parameter count and robustness / alignment in more depth as a valuable direction for future work.

Overall, the Llama model pair results reinforce the pattern seen in the case study and the BERT replication, where distilled students are significantly less robust to component ablation than their teachers, even at substantially larger model sizes. Moreover, the smaller teacher–student robustness gap in this pair helps explain its higher alignment score (Section 4), consistent with the view that more similar influence distributions yield higher alignment under our metric.

### H.2 Question answering: SimpleQA

We additionally extend our analysis to a more complex and realistic downstream NLP task, involving question answering: SimpleQA (Wei et al., 2024). SimpleQA involves the model answering short fact-seeking questions, where there is just one correct answer. This task is effective as it contains significantly less structure than the numeral sequence completion and indirect object identification tasks, which rely on syntactic patterns to obtain the correct answer. We again focus on reproducing our robustness and alignment metric findings, due to the complexity of identifying and verifying individual component differences in this larger model pair. We study 200 randomly sampled examples from the dataset.

In this task, we observe high confidence in the correct answer across the teacher and student models (logit difference: 3.24 and 2.53, respectively), with a relatively high alignment score of 0.9812, consistent with the pattern we observed of similar and high performance coupled with a high alignment score (0.9778) between this model pair in the numeral sequence completion task. We additionally successfully replicate our findings of lowered robustness to component ablation in the student, with 1.89% of components resulting in a performance drop of more than 10% in the student, compared with 0.66% in the teacher (mirroring the numeral sequence task, where we saw 2.27% in the student and 0.95% in the teacher). This provides further evidence that, even on more complex downstream tasks, the trend of the student undergoing shifts towards decreased robustness to component ablation persists.

## I    BERT and DistilBERT replication study (numeral sequence completion)

In this section, we provide details of a complementary study using BERT (Devlin et al., 2019) as the teacher model (12 layers, 12 heads, 109M parameters) and DistilBERT (Sanh et al., 2020) as the student (6 layers, 12 heads, 66M parameters) on the numeral sequence completion task (using 100 randomly-sampled examples). The aim of this study is to evaluate the generalizability of our findings on the GPT2 pair. We do not conduct as thorough of an analysis here, solely focusing on a few key attention heads and MLPs with the same methodology as outlined in Section 3.1.

These BERT models differ from the GPT2 model pair in that they are bidirectional encoder-only architectures rather than autoregressive decoder-only models. This distinction results in different attention patterns, where BERT models attend to both past and future tokens simultaneously, unlike the GPT2 models, which use causal attention to prevent future token leakage. As a result, we expect the internal circuits and restructuring behaviors during distillation to show both parallels and architecture-specific differences to the GPT2 variants.

The BERT model pair also achieves significantly worse performance on the task than the GPT2 models, with the teacher achieving a logit difference of 1.17 and the student achieving 0.49, compared with GPT2 achieving 6.11 and DistilGPT2 achieving 3.75. Naturally, as a result of this performance discrepancy between the two pairs, certain component functionalities in the BERT pair are less-defined and more noisy.

We find that, despite architectural differences, the BERT and DistilBERT models show highly-similar patterns of computational restructuring during distillation to those observed in the GPT2 pair. In particular, DistilBERT tends to compress multiple functionalities into fewer components and shows increased reliance on individual attention heads and MLPs, often resulting in brittle behavior when these are ablated. Additionally, both DistilBERT and DistilGPT2 were observed to discard the "similar member detection" functionality present in their respective teachers, suggesting that such functional deletions may be semi-universal across architectures. However, we also note greater noisiness and less distinct specialization in some components,

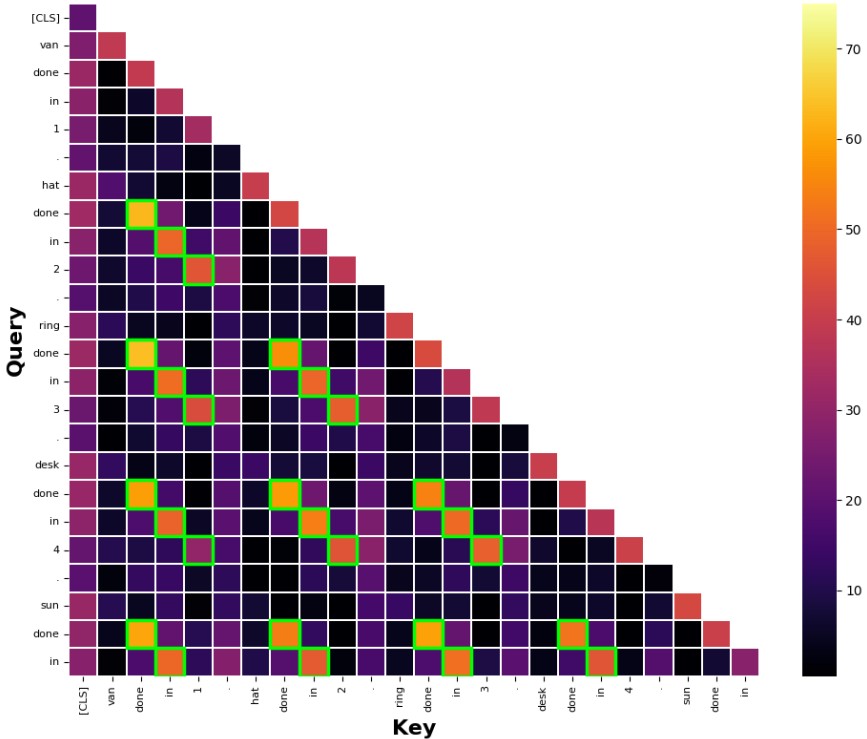

Figure 17: Teacher (T-L1-H11) QK attention matrix for the similar member head functionality

particularly in the student model, likely reflecting the lower overall performance of the BERT models on this task. These findings indicate that the mechanistic trends identified in our main study generalize beyond autoregressive models and specific architectures.

## I.1 Component comparisons

### I.1.1 Attention heads

**Similar member detection (T-L1-H11).** An attention head was found in the teacher model which pays strong attention between each token and previous mentions of that same token (Figure 17). This functionality serves to detect repeated elements, which was also a highly-influential functionality in the GPT2 teacher model, although it was not seen in the DistilGPT2 student model. Interestingly, we could not identify this functionality in DistilBERT either, suggesting that this functionality is not considered a crucial one by student models. Further work looking into the universality of student-discarded functionalities would be interesting here.

**Numeral detection (T-L4-H2 / S-L1-H9).** This functionality is responsible for encoding the numeral sequence through high self-attention between each numeral and its predecessors. The student model is seen to partially copy this functionality across, where there is high self-attention between every numeral and its immediate predecessor, but the range of this self attention does not extend past the previous numeral. In the teacher, high self-attention is seen between each numeral and all predecessors (Figure 18). This partial implementation by the student is likely to be a contributing factor towards the lower task performance seen by the student in the numeral sequence completion task.

When ablated, this head causes a performance change of -6.57% in the teacher, and -100.00% in the student, completely erasing the student's ability to perform this task. The reason for such a high dependence on this head by the student could be due to the fact that it seems to implement a secondary functionality concurrently, where there are diagonals in the QK-matrix of high similarity, potentially indicating encoding

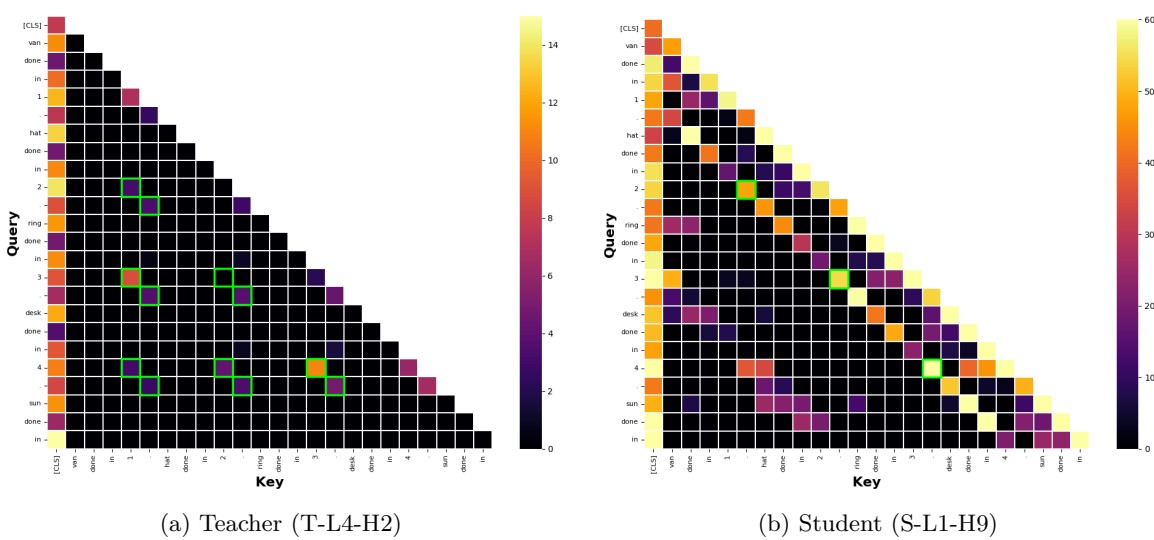

(a) Teacher (T-L4-H2)    (b) Student (S-L1-H9)

Figure 18: Teacher (left) and student (right) QK attention matrices for the numeral detection head functionality

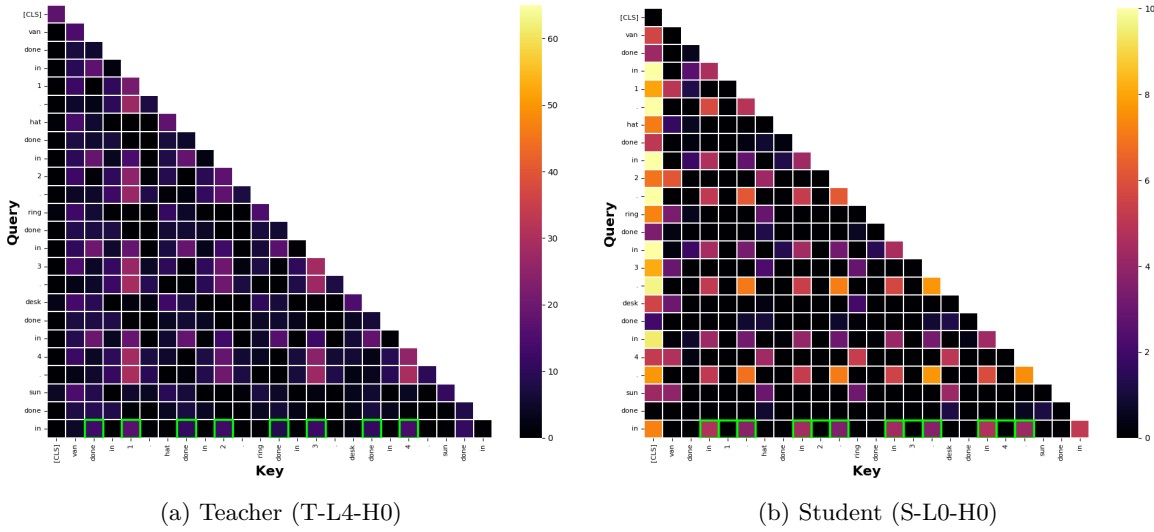

(a) Teacher (T-L4-H0)    (b) Student (S-L0-H0)

Figure 19: Teacher (left) and student (right) QK attention matrices for the numeral mover head functionality

of local context into each token. The teacher head appears much more specialized on the numeral detection task. This compression of multiple functionalities into a single component was also observed in the GPT2 pair, where two MLP components from the teacher were effectively compressed into a single MLP component in the student (Section 3.2.2).

**Numeral mover (T-L4-H0 / S-L0-H0).** We observe that both models contain distinct attention heads responsible for transferring numeral information to the final token's representation (Figure 19). These "numeral mover" heads operate by assigning strong attention from the final token to positions immediately surrounding each numeral, typically the two neighboring tokens. This attention pattern enables the final token to aggregate contextual signals that encode numerical content, effectively integrating that information into its own representation. Notably, the student model closely matches the teacher's QK matrix structure on the final row of this head, indicating a strong replication of this functionality. Ablating this head leads to a performance drop of -10.83% in the teacher and -44.34% in the student, again consistent with the GPT2 study, reinforcing the observation that student models tend to over-rely on retained functional components.

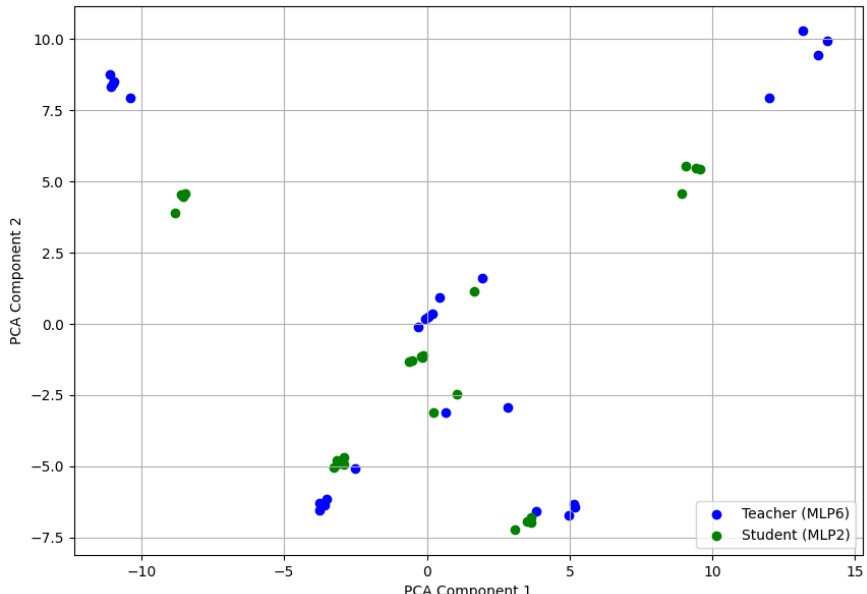

Figure 20: PCA projection of mean token MLP activations within MLP-T-6 and MLP-S-2 for BERT models

### I.1.2 MLPs

**MLP-T-6 / MLP-S-2.** We identify a pair of MLPs that are highly important for the numeral sequence completion task in both models, with ablation leading to a performance change of -84.35% in the teacher and -100.00% in the student (Figure 20). The two MLPs are structurally similar, with a cosine similarity of 0.842 between their token activations, suggesting that they perform the same function across models. Both models appear to rely on these MLPs in similar ways, though the student again shows a higher ablation-induced performance change.

**MLP-T-11 / MLP-S-5.** This MLP pair is anomalous as it indicates the first observed case where the ablation-induced performance change is greater in the teacher (-76.68%) than the student (-4.98%), despite the fact that the activation structure is highly similar (cosine similarity: 0.928) (Figure 21). The large discrepancy in reliance on this MLP is indicative of a differing underlying algorithm to perform the numeral sequence task between the two models, with the student placing its reliance on a different set of MLP functionalities than the teacher.

**First token divergence.** Notably, we do not find evidence of a significant divergence in first token activation between any impactful MLP layers across the BERT models, unlike the pronounced difference seen in the MLP-T-11 / MLP-S-5 pair in the GPT2 study (Section 3). While this does not necessarily imply that the earlier observation was anomalous, it highlights the need for further investigation into this phenomenon to assess whether it is a recurring outcome of the KD process or not, and to understand its implications for model divergence more broadly.

### I.2 Ablation performance changes

Our findings on ablation-induced performance changes in the BERT study (where we ablate individual components and measure the change in logit difference on the numeral sequence completion task relative to the unablated model) are strongly consistent with those observed in the GPT2 / DistilGPT2 study. Once again, we find that the student model places significantly greater reliance on individual components than the teacher, with many ablations leading to complete functional collapse in the student (Figure 22). These results further support the generalizability across different architectures of the trends identified in our main study.

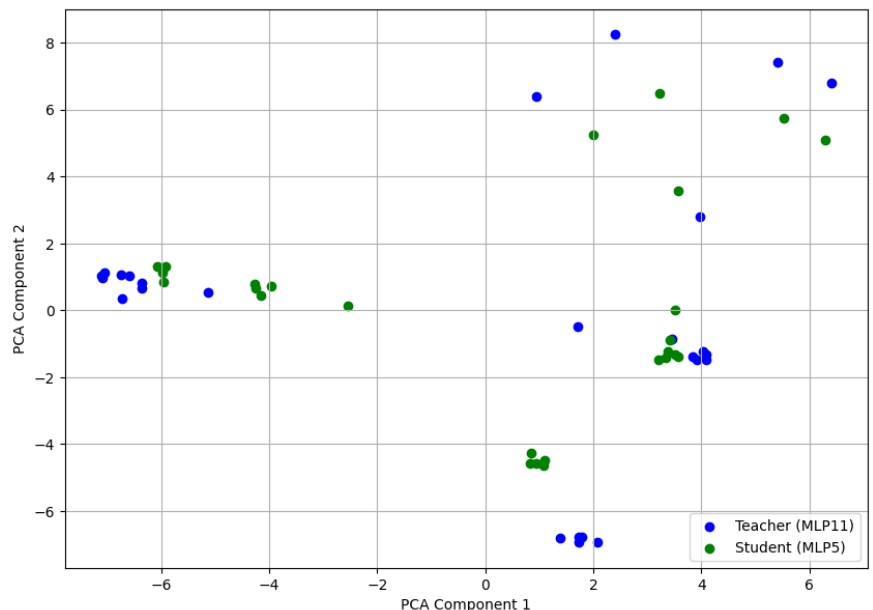

Figure 21: PCA projection of mean token MLP activations within MLP-T-11 and MLP-S-5 for BERT models

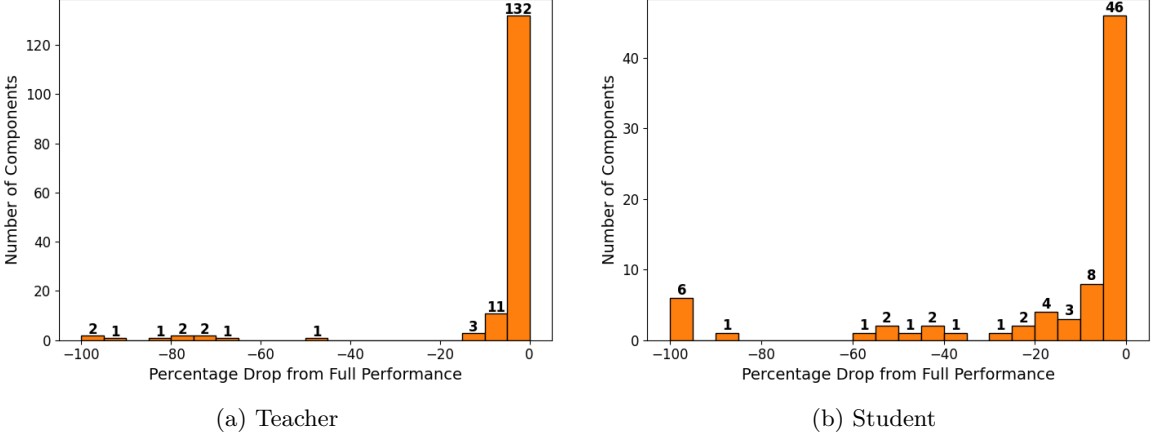

(a) Teacher

(b) Student

Figure 22: Distribution of performance drops caused by ablation across components in both the teacher (left) and student (right) BERT models

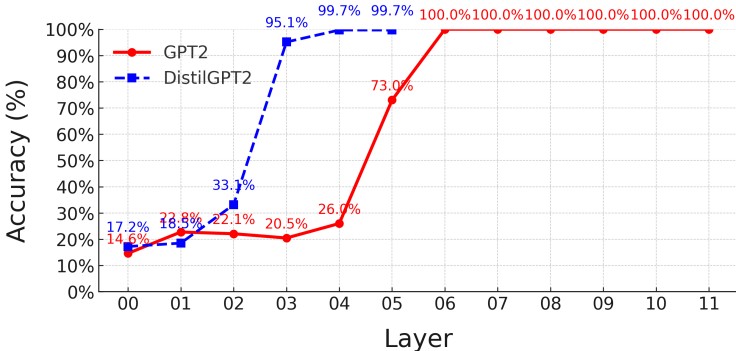

Figure 23: Probe accuracy across layers for GPT2 and DistilGPT2 on prediction of the $i$-th element of the numeral sequence

## J  Role validation details and results

Here, we provide further details regarding the methodology and results of our role validation techniques, broken down by key attention head components across the GPT2 and DistilGPT2 model pair for the numeral sequence completion task.

For training of linear probes, we extract token-level feature vectors from the model's intermediate activations for each prompt and layer (either the residual stream post-attention, pre-MLP, or the value vectors of a specified attention head). For classification tasks (e.g., numeral prediction), we train a single linear layer with cross-entropy using Adam ($\text{lr} = 1e - 3$) for 20 epochs on an 80/20 split; for multi-label tests (e.g., position-memory) we use logistic loss and report AUROC. We repeat this per layer to produce layer-wise curves.

### J.1  Numeral detection (T-L4-H4 / S-L2-H4)

#### J.1.1  Activation patching

We find that logit difference recovery is entirely concentrated in the attention blocks of layer 4 when patching the numerals of the sequence, yielding a logit difference improvement of 0.411 in the teacher and 0.491 in the student. This translates to a $\exp(0.411) \approx 1.51\text{x}$ and $1.63\text{x}$ increase in the clean prompt's correct to incorrect token probability ratio, respectively. This is causal evidence that this layer is solely responsible and sufficient for encoding the numeral sequence for the remainder of the circuit. To pinpoint which heads are responsible, we patch the QK circuits of each head in layer 4. We find that recovered performance is heavily concentrated in TL4H4, TL4H10 (likely a backup), and SL2H4. This provides targeted causal evidence that these heads actively encode numeral structure, not just attend to it.

#### J.1.2  Probing

We train single-layer linear probes to predict the $i$-th numeral in the sequence from the residual stream at each layer (Figure 23). Accuracy is low across early layers, then jumps to $\sim 73\%/\sim 95\%$ at layer 5/3, and reaches perfect decodability by layer 6/4 onwards. This shows that numeral information becomes linearly accessible in the layer directly after these components, consistent with attention heads in that layer writing this information into the residual stream.

### J.2  Numeral mover (T-L7-H11 / S-L3-H11)

#### J.2.1  Activation patching

We ran an activation patching experiment to test whether these heads move task-relevant information (the sequence) to the final token. Patching in clean activations at the final token and measuring logit difference

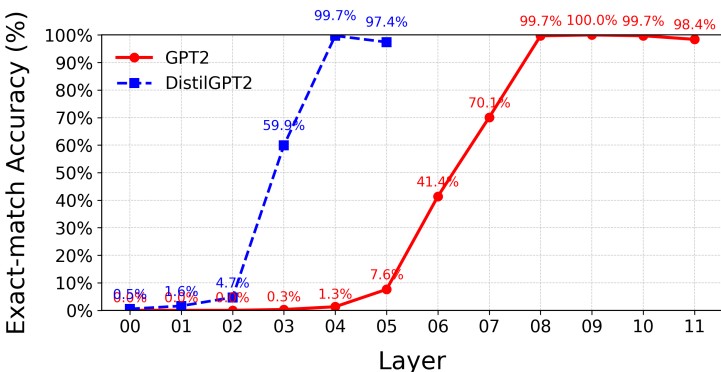

Figure 24: Probe accuracy across layers for GPT2 and DistilGPT2 on prediction of the full numeral sequence

recovery shows peaks in layer 7 (teacher, 3.53× layer mean) and layer 3 (student, 3.42× layer mean). Analyzing OV circuits, T-L7-H11 achieves a +0.164 margin (z = 3.6), recovering 8.3x more than the next best in its layer. SL3H11 reaches +0.344 (z = 3.3), with 6.36x more in its layer. These results causally implicate these heads as key movers of the numeral sequence.

### J.2.2 Probing

Probing for the full numeral list from the final-token residual stream (Figure 24), we obtain accuracy which jumps from 7.60% in layer 5 to 41.4% in layer 6 and 70.1% in layer 7 in the teacher, and from 4.7% in layer 2 to 59.9% in layer 3 in the student. These numbers imply that T-L7-H11 and S-L3-H11 are writing information to the residual stream such that the probe is able to linearly decode the sequence in the subsequent layer. Accuracy remains high in later layers, suggesting the sequence is preserved for downstream use.

### J.3 Successor computation (T-L9-H1 / S-L4-H1)

### J.3.1 Activation patching

Following Gould et al. (2024), we project each head's OV contribution at the final token into vocabulary space and compute two metrics: (i) a successor score (the percentage of examples where the ground-truth next token appears in the head's top-5 logits); and (ii) a copy score (the percentage where the last-given token reappears in the head's top-5 logits). We find that T-L9-H1 attains a successor score of 87.37% and a copy score of 59.27% (vs. a model-wide head average successor score of 3.29%). S-L4-H1 yields 96.64% successor and 3.61% copy (vs. 4.89% average successor). This sharp specialization, especially in the student, supports a focused successor head role.

### J.3.2 Probing

Probes for predicting the next numeral (Figure 25) show teacher accuracy spikes from 29.4% in layer 7 to 75.3% in layer 8 and 94.3% in layer 9, then drops back down for the remainder of the layers. The student shows a similar peak in accuracy at layer 4 of 77.6%. These alignments reinforce the role of T-L9-H1 / S-L4-H1 in encoding the correct successor.

### J.4 Similar member detection (T-L1-H5)

### J.4.1 Probing

Although this attention head is referred to as the "similar member detection" head in prior work (Lan et al., 2024), we test whether this component is primarily encoding token repetition, or simply just focusing on encoding the previous numeral for each point in the sequence. A linear classifier for the previous numeral at

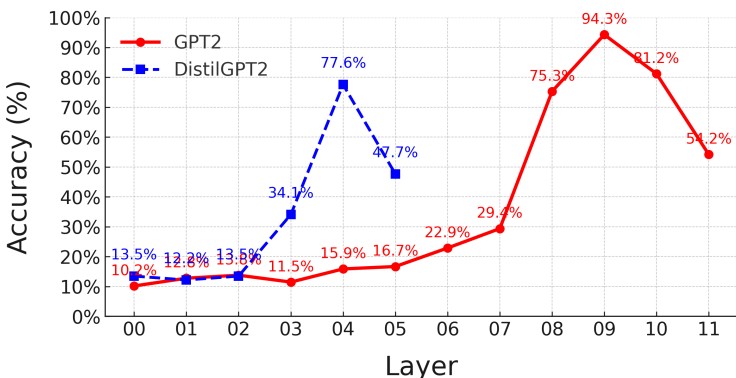

Figure 25: Probe accuracy across layers for GPT2 and DistilGPT2 on prediction of the next numeral

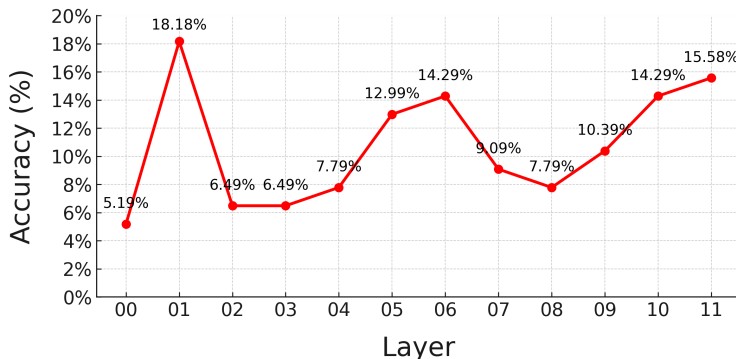

Figure 26: Probe accuracy across layers for GPT2 and DistilGPT2 on prediction of the previous element at each position of the numeral sequence

each position of the sequence (Figure 26) on head values showed peak 18.18% accuracy at T-L1-H5 (vs 5.19% at L0), dropping to 6.49% in layer 2, with the next-highest peak of 15.58% in layer 11. This was not seen in the student, with accuracy peaking in the last layers at 16.88% in layer 4. Lower scores are expected here due to single-head probing.

When instead training a binary probe to predict prior token occurrence, we see low accuracy across layer 1 (56.5%), with performance of the probe peaking at layer 7 (82.2%). This is supporting evidence for the fact that the primary goal of T-L1-H5 is not to track token repetition, but instead to encode the previous numeral. We do however keep the same role title throughout the rest of this paper to maintain consistency and to avoid confusion.

### J.4.2 Activation patching

Activation patching over the OV circuit confirmed its causal role, with T-L1-H5 showing the largest recovery in its layer (2.84x next best; z = 2.93).

## K   Robustness quantification

To quantify robustness between each studied model on the numeral sequence completion task, we report the mean (95% bootstrap CIs) values of the component ablation-induced drops in performance in Table 2. This number captures the degree of which the model's performance drops under component ablation, where higher values represent less robustness, and lower values more robustness. We can see that the Llama model pair

shows both significantly higher mean robustness than the other smaller pairs, as well as significantly lower difference in robustness.

## L  Relationship between model compression and robustness

We provide additional information on quantification of the relationship between model compression (in terms of parameter count) and robustness (i.e., ablation-induced performance drop). This is an important trade-off to consider, as it has broad implications for decisions on the parameter count of the student model, which is often very domain-specific. For example, if aiming to distill a student for low-resource environments, it may be more acceptable to sacrifice some robustness for higher compression, but this may not be the case for more critical domains.

To quantify this relationship, we ablate each attention head and MLP across tasks for three teacher–student pairs (GPT2 124M→82M, BERT 109M→66M, Llama 8B→4B) and measure the resulting drop in logit-difference between the correct and incorrect token. We find that compression consistently increases component-level brittleness. For GPT-2 ($C = 33.9\%$), the student exhibits a much larger mean drop than the teacher (12.24pp vs. 3.06pp; $\Delta = 9.18$pp). This corresponds to $\beta_{\mathrm{mean}} = 26.94$pp$\cdot C^{-1}$, i.e. 2.69pp per $0.1, C$ (95% CI: 0.92–4.65 per $0.1, C$). BERT shows a similar pattern: at $C = 39.4\%$, the student drop is 16.89pp vs. 6.26pp in the teacher ($\Delta = 10.62$pp), giving $\beta_{\mathrm{mean}} = 26.75$pp$\cdot C^{-1}$ (2.68pp per $0.1, C$; 95% CI: 0.94–4.56). Llama ($C = 50\%$) is less dramatic but still consistent with increased brittleness under compression: 2.20pp vs. 0.84pp ($\Delta = 1.36$pp), with $\beta_{\mathrm{mean}} = 2.72$pp$\cdot C^{-1}$ (0.27pp per $0.1, C$; 95% CI: 0.14–0.43).

Across architectures, this implies an increase in brittleness of **0.27–2.69pp per** $0.1, C$, showing that across our six models, parameter compression provides a clear correlation with decreased ablation-induced robustness. We highlight confirming these findings across many more diverse teacher-student pairs (e.g., differing distillation loss functions, architectural differences, parameter sizes, etc.) as an interesting direction for future work.

## M  Alignment metric design choice sensitivity analysis

We evaluate the sensitivity of the alignment metric (Section 4) to two key design choices: (i) how component influences are normalized before comparing teacher and student, and (ii) how teacher and student components are matched. We run a sweep over influence normalization (max-normalization as used in the main text, as well as $\ell_1$ and $\ell_2$ normalization) and over matching (greedy nearest-neighbor baseline, one-to-one Hungarian matching, and a soft top-$k$ assignment with $k=5$ and temperature $T=1$). We conduct this analysis on the numeral sequence completion task, which is the only task studied across all three model pairs in our experiments, enabling a consistent model-wise comparison.

Table 5 reports the resulting alignment scores. Across all eight variants, the induced ranking of model-pair alignment scores is unchanged relative to the original cross-model findings at N=100: The Llama pair remains highest, GPT remains intermediate, and BERT remains lowest (Spearman $\rho \approx 1.0$ and pairwise order agreement $= 1.0$ versus baseline for all variants, computed over the three model pairs). Absolute scores were observed to shift modestly depending on the variant (mean $|\Delta| \approx 0.02$ - $0.05$ across variants; worst-case $|\Delta| \leq 0.083$ in this sweep), but these changes do not affect any ranking-based qualitative conclusions drawn from the metric.

## N  Circuit extraction threshold sweep

To address robustness of our findings in Section 3.2, we add a sweep over the circuit extraction threshold $T_n \in 0.10, 0.15, 0.20, 0.25, 0.30$ and report: (i) circuit size (nodes/edges), and (ii) completeness and faithfulness as functions of $T_n$. We observe the expected smooth pattern, where circuit size is highly stable (28-30 nodes total; 18-19 heads and 10-11 MLPs), while completeness decreases monotonically as $T_n$ increases (91.9%, 85.1%, 80.1%, 75.8%, 70.7% of baseline logit-difference retained when keeping only the circuit). Faithfulness is essentially unchanged across thresholds, as ablating the extracted circuit consistently collapses performance (faithfulness $\approx 4\%$ of baseline logit-difference for all $T_n$), indicating that the same core causal mechanism is captured throughout the sweep.

| Variant | GPT2 & DistilGPT2 | BERT & DistilBERT | Llama & Minitron |
|---|---|---|---|
| max + greedy (baseline) | 0.865 | 0.809 | 0.938 |
| max + hungarian | 0.893 | 0.835 | 0.943 |
| max + soft top-$k$ ($k{=}5, T{=}1$) | 0.812 | 0.756 | 0.905 |
| $\ell_1$ + greedy | 0.937 | 0.831 | 0.966 |
| $\ell_1$ + hungarian | 0.948 | 0.852 | 0.966 |
| $\ell_1$ + soft top-$k$ ($k{=}5, T{=}1$) | 0.893 | 0.769 | 0.956 |
| $\ell_2$ + greedy | 0.918 | 0.823 | 0.960 |
| $\ell_2$ + hungarian | 0.924 | 0.844 | 0.955 |
| $\ell_2$ + soft top-$k$ ($k{=}5, T{=}1$) | 0.875 | 0.764 | 0.946 |

Table 5: Sensitivity of alignment scores (N=100) on the numeral sequence completion task to influence normalization and component matching choices. All variants preserve the same ranking over model pairs.

