# OpenReview forum: "Distilled Circuits: A Mechanistic Study of Internal Restructuring in Knowledge Distillation"
_TMLR — Accepted by TMLR_

### Review · Reviewer_yYWH · 2025-12-24

**Summary Of Contributions:**

In this article, the authors present a mechanistic interpretability study of knowledge distillation, analyzing how internal circuits, representations, and component roles change when compressing a large teacher model into a smaller student. The model GPT-2 and its distilled version, DistilGPT-2, are used as the primary case study, but the analysis is also extended to BERT/DistilBERT and Llama/Minitron. It is shown that distilled students often compress, merge, or discard internal components, leading to increased reliance on fewer critical heads and MLPs. This restructuring preserves surface-level performance but reduces robustness to ablations and distribution shifts.Beyond qualitative circuit analysis, the paper introduces an influence-weighted alignment metric that aims to quantify functional similarity between teacher and student internal mechanisms, going beyond output-level metrics.

Strengths:
- The paper is clearly written and well structured. The study is conducted thoroughly and with considerable care.

- The authors address an interesting and timely problem, namely understanding what is actually preserved and altered internally during knowledge distillation. By focusing on the mechanistic changes induced by distillation rather than solely on output behavior, the work targets an important gap in the current literature.

Weaknesses:
- It is unclear to what extent the results generalize beyond the relatively simple, highly structured tasks studied in the paper, such as numeric sequence completion. These tasks admit compact and well-defined circuits, which may exaggerate the apparent degree of component compression and brittleness observed in distilled models. It would therefore be important to clarify whether similar mechanistic restructuring and robustness trade-offs arise in more complex, naturalistic tasks.

- The circuit extraction and subsequent conclusions rely heavily on the choice of the threshold $T_n=0.2$, yet the justification for this specific value is not clearly stated. An analysis of how the results change under different threshold values (e.g., lower or higher ablation cutoffs) would be important to establish the robustness of the findings.

- the paper presents a well-executed mechanistic study that provides insights into how knowledge distillation reshapes internal computation. However, the broader generality and practical importance of the results remain somewhat unclear, given the limited task diversity and the reliance on highly structured settings. I had hoped to see a stronger theoretical analysis; however, the paper offers only limited contributions in this direction. As a result, it is hard to judge how impactful these findings will be for understanding distillation in more complex or realistic scenarios.

**Audience:**

Yes

**Audience Explanation:**

I would not say “no” here, as there will likely be some interested members of the audience, but I expect the overall appeal to be somewhat limited. In particular, the lack of theoretical results and the resulting uncertainty about generalization across tasks (rather than across models) may restrict broader interest.

Update: The additional studies strengthen the claim on the generalization across tasks

**Claims And Evidence:**

Yes

**Claims Explanation:**

The empirical and mechanistic analyses presented in the paper are generally careful and clearly executed, and they provide convincing evidence for the specific claims made within the studied tasks and model pairs.

**Requested Changes:**

I wish the authors had provided more theoretical insights here rather than focusing solely on the quantitative study, though I understand that doing so would significantly shift the paper's focus. If that is not feasible, the task design would require more careful engineering (even though I don't know how its possible to define representative tasks)

---

> ### Author Response · Authors · 2026-01-20
> **Official Comment by Authors**
>
> Thank you for your excellent comments and suggestions! Below, we go through each one:
>
> > It is unclear to what extent the results generalize beyond the relatively simple, highly structured tasks studied in the paper ..., which may exaggerate the apparent degree of component compression and brittleness observed in distilled models ...
>
> We agree that broader natural-language evaluation would strengthen the paper. Beyond our existing IOI-style natural-language results (Appendix G), we added a more realistic downstream question-answering task (SimpleQA [1]) on Llama-3.1-8B (teacher) and Minitron-4B (student).
>
> On this task. we see that both models perform well (logit diff 3.24 vs 2.53) and remain highly aligned ($A$ = 0.9812), consistent with the numeral-sequence task for this pair (A = 0.9778). The student is again less robust to component ablation: 1.89% of components cause a >10% performance drop vs 0.66% for the teacher (numeral sequence: 2.27% vs 0.95%).
>
> Please note that a full Section 3 style replication is out of scope here because identifying, validating, and comparing individual components from the circuit becomes substantially more complex and time-consuming with the increased number of components and layers present in these models
>
> ### Change: Updated the manuscript with the new SimpleQA task and results.
>
> > The circuit extraction and subsequent conclusions rely heavily on the choice of the threshold $T_n$ = 0.2 ...
>
> We clarify that $T_n$ is used only for circuit extraction and Section 3’s circuit-based analyses (completeness/faithfulness/minimality). In contrast, the alignment metric is computed over all components using continuous influence scores and does not depend on circuit membership or $T_n$. We also wish to note that this task was chosen as it has previously been studied on the GPT2 model by Lan et al. (2024), motivating our choice of 0.2 to replicate their hyperparameters for simplicity.
>
> To address robustness, we also added a sweep over $T_n \in$ {0.10, 0.15, 0.20, 0.25, 0.30}. Circuit size is stable (28-30 nodes: 18-19 heads, 10-11 MLPs). Completeness decreases smoothly with $T_n$ (91.9%, 85.1%, 80.1%, 75.8%, 70.7% of baseline logit diff retained), while faithfulness is essentially unchanged: ablating the extracted circuit collapses performance to $\approx$4% of baseline for all $T_n$, indicating the same core causal mechanism is captured throughout.
>
> ### Change: Updated the manuscript with an appendix (N) showing the $T_n$ sweep findings, justify our chosen threshold of $T_n$ = 0.2.
>
> >... I had hoped to see a stronger theoretical analysis; however, the paper offers only limited contributions in this direction. As a result, it is hard to judge how impactful these findings will be for understanding distillation in more complex or realistic scenarios.
>
> >I wish the authors had provided more theoretical insights here rather than focusing solely on the quantitative study, though I understand that doing so would significantly shift the paper's focus. If that is not feasible, the task design would require more careful engineering (even though I don't know how its possible to define representative tasks)
>
> As per the above response, we have since updated our work to include a more complex and natural task (SimpleQA), which we hope will improve the impact of our work.
>
> To improve the clarity of the broader generality and practical importance of our results (and to address reviewer 8qLZ’s request for a practitioner-focused guide for use), we have revised the manuscript to include a short practitioner-focused subsection (4.3, “Using the alignment metric in practice”). It explains both when to use the metric and adds guidance on interpretation, using $A$ to choose among students with similar accuracy (noting that performance gaps alone can be misleading) and reporting $A$ alongside a simple robustness summary, since students may rely more heavily on a smaller set of components.
>
> As for theoretical insights, we do think that this would significantly shift our work’s focus, as this work is primarily concerned with demonstrating that student models’ internal representations, computational circuits, and robustness to component ablation can differ significantly to their teachers, and that we can automatically quantify this with an alignment metric. However, we do see the benefit of this style of analysis, and have updated Section 5.1 to discuss how theoretical insights could lead to a better understanding of when and why knowledge distillation preserves, merges, or re-routes internal computations, and how these capacity-driven changes relate to redundancy and robustness. We hope that our work may serve as motivation for further progress in this direction.
>
> ### Change: Expanded Section 5.1 (future theoretical directions) and added Section 4.3
>
> ## References:
> [1] Wei, J., et al. (2024) "Measuring short-form factuality in large language models." arXiv preprint arXiv:2411.04368.

---

> > ### Comment · Reviewer_yYWH · 2026-02-16
> >
> > I would like to thank the authors for their answer. The additional studies strengthen the claim on the generalization across tasks. I have no further requests and think that the article should be accepted.

---

### Review · Reviewer_8qLZ · 2025-12-30

**Summary Of Contributions:**

**Summary:**

The paper investigates how knowledge distillation alters internal computation in neural networks, using mechanistic interpretability tools to compare teacher–student model pairs. Through detailed circuit-level analyses on sequence completion and indirect object identification tasks, the authors show that distilled students often compress, merge, or discard internal components, leading to increased reliance on fewer mechanisms and reduced robustness. The work also introduces an influence-weighted alignment metric intended to quantify functional similarity between teacher and student beyond output behavior, and validates it across tasks and architectures (GPT2, BERT, Llama families).

**Strengths:**

* S1: The paper adopts a mechanistic perspective by moving beyond surface-level behavioral comparisons and instead directly analyzing the internal circuits of teacher and student models using established mechanistic interpretability techniques such as component ablation, activation patching, and path-specific interventions, which enable a much deeper understanding of how knowledge distillation reshapes internal computation.

* S2: The experimental framework for circuit discovery is carefully structured and quantitatively grounded, with explicit enforcement of completeness, faithfulness, and minimality through clearly defined ablation thresholds, making the identified circuits both necessary and sufficient for task performance and reducing the risk of arbitrary or overinclusive interpretations.

* S3: The proposed influence-weighted alignment metric represents a meaningful methodological advance, as it explicitly incorporates task-specific component importance into the measurement of teacher–student similarity and therefore captures functional alignment in a way that global representational similarity metrics fail to do, particularly in cases where output behavior appears similar despite internal divergence.

* S4: The inclusion of multiple teacher–student pairs across different architectures and scales, including GPT, BERT-style, and Llama-based models, provides convincing evidence that the observed patterns of component compression, reorganization, and increased reliance on fewer mechanisms are not isolated to a single model family.

* S5: The empirical findings offer clear and interpretable insights into the consequences of knowledge distillation, showing that student models often become more brittle by concentrating critical functionality into fewer components, which in turn helps explain previously observed reductions in robustness and increased sensitivity to distribution shifts.

**Weaknesses:**

* W1: The analysis is largely centered on relatively simple and highly controlled tasks, such as sequence completion and indirect object identification, which are well-suited for mechanistic study but make it difficult to assess whether the same internal restructuring and robustness trade-offs would persist for more complex, real-world natural language processing tasks.

* W2: Several steps in the component matching and functional role attribution process rely on manual inspection and qualitative judgment, which may limit the scalability of the approach to larger models or broader task suites and could pose challenges for reproducibility across different research groups.

* W3: The experimental evaluations are conducted on relatively small datasets, typically involving around 100 examples per task, which may be sufficient for qualitative analysis but raises concerns about the statistical stability and generalizability of quantitative measures such as alignment scores and ablation-induced performance drops.

* W4: The alignment metric, while intuitively motivated and empirically validated, depends on specific design choices such as influence normalization, similarity thresholds, and greedy matching procedures, and the paper does not provide extensive ablation studies to demonstrate that the conclusions are robust to alternative reasonable design decisions.

* W5: Although the ablation experiments convincingly demonstrate increased sensitivity of student models to component removal, the paper sometimes extrapolates these results to broader claims about downstream failures and deployment risks, and these connections remain largely speculative without direct empirical validation on applied or safety-critical settings.

**Audience:**

Yes

**Audience Explanation:**

Researchers in mechanistic interpretability, model compression, and robustness would likely find the results and methodology valuable, particularly the circuit-level perspective on distillation and the proposed alignment metric.

**Broader Impact Concerns:**

The work highlights potential safety and robustness risks of deploying distilled models that appear behaviorally competent but rely on brittle internal mechanisms. While primarily analytical, the findings could inform safer evaluation and selection of compressed models. No direct negative impacts are apparent; concerns are largely about misuse through overconfidence in distilled systems.

**Claims And Evidence:**

Yes

**Claims Explanation:**

The central claims that knowledge distillation preserves broad functionality while significantly restructuring internal computation and reducing robustness are well supported by systematic circuit ablations, comparative analyses, and consistent patterns across multiple model families. The alignment metric is validated through controlled noise injection and cross-task comparisons, lending credibility to its intended purpose. However, the evidence is strongest for small-to-mid-scale models and relatively narrow tasks, so claims about general robustness and deployment implications should be interpreted with appropriate caution.

**Requested Changes:**

1. The paper would benefit from evaluating at least one more realistic downstream NLP task, such as question answering or summarization, to assess whether the observed mechanistic restructuring and robustness effects extend beyond controlled, synthetic settings.

2. Increasing dataset sizes or providing additional variance and confidence analyses would strengthen the statistical reliability of the reported alignment scores, ablation effects, and cross-model comparisons.

3. The authors should include ablation studies or sensitivity analyses for the key design choices underlying the alignment metric, such as influence normalization and component matching, to demonstrate that the results are robust to reasonable alternatives.

4. Clarifying how the current manual component matching and role attribution procedures could be automated or scaled would improve the practicality and reproducibility of the approach for larger models and broader analyses.

5. A more explicit discussion of the limitations of using ablation-based robustness as a proxy for real-world robustness would help contextualize the findings and avoid overinterpretation of the results.

6. The paper would be strengthened by providing clearer guidance on how practitioners could apply the proposed alignment metric in practice, for example when selecting or evaluating distilled models for deployment.

7. The authors should more clearly distinguish which conclusions are specific to the tasks studied and which are expected to generalize across tasks, architectures, or model scales.

8. Improving clarity around the computational cost, scalability, and feasibility of the proposed analyses for larger models would help readers assess the practicality of applying these methods in real-world research or deployment settings.

---

> ### Author Response · Authors · 2026-01-20
> **Official Comment by Authors**
>
> Thank you for your excellent comments and suggestions! Below, we go through each one:
>
> > The paper would benefit from evaluating at least one more realistic downstream NLP task ...
>
> We agree that broader natural-language evaluation would strengthen the paper. Beyond our existing IOI-style natural-language results (Appendix G), we added a more realistic downstream question-answering task (SimpleQA [1]) on Llama-3.1-8B (teacher) and Minitron-4B (student).
>
> On this task, both models perform well (logit diff 3.24 vs 2.53) and remain highly aligned ($A$ = 0.9812), consistent with the numeral-sequence task for this pair ($A$ = 0.9778). The student is again less robust to component ablation, with 1.89% of components causing a >10% performance drop vs 0.66% for the teacher (numeral sequence: 2.27% vs 0.95%).
>
> A full Section 3-style component-level replication on these larger models is out of scope here, since identifying, validating, and comparing individual components becomes substantially more complex with many more layers/components.
>
> ### Change: Added SimpleQA setup + results.
>
> > Increasing dataset sizes or providing additional variance and confidence analyses ...
>
> ### Change: Increased dataset sizes (N=384 for numeral and word sequence completion, due to dataset size limit, N=500 for IOI, N=200 for simpleQA) and included 95% CIs in alignment metric results table.
>
> > The authors should include ablation studies or sensitivity analyses for the key design choices underlying the alignment metric ...
>
> We tested robustness to two key choices on numeral-sequence completion: (i) influence normalization (max vs L1 vs L2) and (ii) component matching (greedy NN baseline vs Hungarian vs soft top-k assignment). Across all variants, the relative ranking of model-pair alignment scores is unchanged. Absolute scores vary modestly (mean $|\delta| \approx 0.02-0.05$; worst-case $\leq$ 0.083), but these shifts do not affect ranking-based conclusions and are expected under changes in influence normalization.
>
> ### Change: Added full table + summary statistics (App. M).
>
> > Clarifying how the current manual component matching and role attribution procedures could be automated or scaled would improve ...
>
> Automatic component matching used in the metric is already described in Sec. 4.1.2. Role attribution currently involves a manual hypothesis step (e.g., via QK-matrix inspection) followed by validation via activation patching and linear probing. As future work, we note that interpretability agents [1, 2] may help scale this hypothesis generation step.
>
> ### Change: Expanded limitations/future work on automation.
>
> > A more explicit discussion of the limitations of using ablation-based robustness as a proxy ...
>
> Thank you for this suggestion. In the revision, we have made it clearer that our metric captures robustness to internal component removal, which is an established causal-intervention tool for assessing component importance, but is only an imperfect proxy for deployment-relevant robustness.
>
> ### Change: Updated limitations/future work and adjusted framing to avoid overinterpretation.
>
> > The paper would be strengthened by providing clearer guidance on how practitioners could apply the proposed alignment metric ...
>
> We agree, and have now added Sec. 4.3 ("Using the alignment metric in practice") which covers: when to use $A$, how to interpret it, and why reporting $A$ alongside a simple robustness summary is useful.
>
> ### Change: Add practitioner-focused subsection (Sec. 4.3).
>
> > The authors should more clearly distinguish which conclusions are specific to the tasks studied ...
>
> We now state explicitly that Sec. 3 findings are specific to the numeral-sequence task and the GPT-2 / DistilGPT-2 pair, while we observe similar (not identical) restructuring/over-reliance patterns on another task (App. G) and other model pairs (Apps. H/I). We also revise the interpretation at the end of Sec. 4.2.2 and the conclusion accordingly.
>
> ### Change: Clarifications in Sec. 3.2, Sec. 4.2.2, and conclusion.
>
> >Improving clarity around the computational cost ...
>
> Thank you for this suggestion. We have since revised the manuscript in multiple places to make this aspect clearer:
>
> ### Change: Add a short discussion of computational cost of the component ablation and path-patching methods in Section 3.1.
>
> ### Change: Add a short discussion of computational cost of the alignment metric in Section 4.1.
>
> ### Change: Add a note in newly-added Section 4.3 “Using the alignment metric in practice”, noting that, in practice, discarding low-influence components in the alignment metric calculation may be useful to combat quadratic scaling of computational cost with larger models.
>
> ## References:
> [1] Schwettmann, S., et al. (2023). Find: A function description benchmark for evaluating interpretability methods. Advances in Neural Information Processing Systems, 2023.
>
> [2] Rott Shaham, T., et al. (2024). A Multimodal Automated Interpretability Agent. ICML 2024.

---

### Review · Reviewer_dz6k · 2026-01-09

**Summary Of Contributions:**

The paper uses mechanistic interpretability methods to compare what a teacher model and a distilled student model are doing inside, focusing on GPT2 vs. DistilGPT2 and also checking BERT/DistilBERT and Llama/Minitron. It finds that distillation can reorganize, compress, or remove teacher components, and the student often relies on fewer circuits than the teacher. It introduces an alignment metric that goes beyond matching outputs by matching components and weighting them by how important each component is for the task. It validates the proposed alignment metric across tasks and argues that similar performance can still hide big differences in internal computation, with implications for robustness and generalization.

**Audience:**

Yes

**Audience Explanation:**

I think the problem this paper considered is on understanding internal circuits of models during model distillation, which is a practical and interesting setup. The findings of this paper should help us understand better on what happens behind the distillation process.

**Broader Impact Concerns:**

I do not have broader impact concerns on this paper.

**Claims And Evidence:**

Yes

**Claims Explanation:**

1. Generality beyond GPT-2/DistilGPT-2: Section 3 provides a thorough analysis of internal circuits in GPT-2 and DistilGPT-2. However, it is unclear how general these findings are. It would strengthen the paper to include additional experiments on other distilled model pairs (beyond brief checks), or to more clearly state which observations are expected to generalize and why.
2. In section 3, the authors did a case study on two tasks to understand the circuits in GPT2 and distillGPT2. Can the authors discuss more on the contributions of this part? Is it new findings mostly on the circuit in distilled models? In terms of the MI approach itself, what are the relevance to existing works?
3. I had trouble understanding the motivation for Section 4 and how it follows from Section 3. The paper would read more cohesively if the authors more explicitly explain: 1) what question Section 4 answers that Section 3 cannot, and 2) how the new metric/analysis in Section 4 is informed by (or validated against) the circuit-level findings in Section 3. Right now the transition feels somewhat abrupt.
4. It would be good to discuss the implications of the results on model training, e.g., distillation. Can the findings in this paper help us design more efficient and better distillation algorithms?
5. Role of training data in distillation: distillation outcomes depend strongly on the data used (distribution, diversity, curriculum, domain). The paper would benefit from discussion—or small experiments—connecting the findings to the distillation data. For instance, would different data mixtures change which circuits are preserved vs. reorganized? Clarifying this would help interpret the results and understand their scope.

**Requested Changes:**

1. For section 2.3, can the authors discuss the motivation for selecting these tasks? What capabilities are we trying to understand from these two tasks? It would be also good to consider other capabilities, like in-context learning, induction heads, arithmetics.
2. For section 4.1, can the authors define the proposed metric formally? For example, you can define the variables and describe how to compute the metric in details. Currently it seems too brief.
3. Can the authors add some discussions on how distillation was used to construct DistillGPT2 and Minitron Llama? For example, this would include how the distilled models are trained and what data is used during this process.

---

> ### Author Response · Authors · 2026-01-21
> **Official Comment by Authors**
>
> Thank you for your excellent comments and suggestions! Below, we go through each one:
>
> > Generality beyond GPT-2/DistilGPT-2: ... It would strengthen the paper to include additional experiments on other distilled model pairs ... or to more clearly state which observations are expected to generalize and why.
>
> We have now updated the manuscript to improve clarity around which observations are expected to generalise and which are task- or model-specific:
>
> ### Change: update Section 3.2 to clarify that the findings here are specific to the numeral sequence task and GPT2 / DistilGPT2 model pair, although we observed similar (but not exact) patterns of internal restructuring and increased component reliance across a different task (Appendix G) and different model pairs (Appendices I and H).
>
> ### Change: alter wording at end of Section 4.2.2 to emphasise that while the scores reported in the paper themselves are not generalisable or model-agnostic, the underlying procedure is
>
> ### Change: add similar clarification in Sec. 5
>
> > In section 3, ... Can the authors discuss more on the contributions of this part? ...
>
> Thank you for flagging a point where clarity can be improved! In Section 3, we extend Lan et al.’s [1] circuit analysis of the numeral sequence completion task on GPT-2 to the DistilGPT student model. We also incorporate a new probe-based validation to verify the functional roles of the circuit’s components, and we analyse differences between the two models’ circuits in both their representations and components. Our circuit-discovery approach (particularly path patching) is heavily inspired by Conmy et al. [2].
>
> ### Change: Make our contributions clearer in Sec. 3
>
> > I had trouble understanding the motivation for Section 4 and how it follows from Section 3 ...
>
> We have now updated the first paragraph of Section 4 to make this link clearer. In summary, the case study in Sec. 3 is extremely time-consuming and intractable for larger models and/or more complex circuitry. Sec. 4 therefore focuses on automating this comparison of functional alignment.
>
> ### Change: Modify first paragraph of Section 4 to improve cohesion.
>
> > It would be good to discuss the implications of the results on model training, e.g., distillation ...
>
> A way that we see our work aiding the distillation training process is via a loss term to measure internal functional alignment. This would be particularly helpful in domains where the student should be discouraged from implementing (potentially hazardous) computational shortcuts.
>
> ### Change: Update Sec. 5.1 to mention exploration of the use of our alignment metric during training.
>
> > ... distillation outcomes depend strongly on the data used ... Clarifying this would help interpret the results ...
>
> > Can the authors add some discussions on how distillation was used to construct DistillGPT2 and Minitron Llama? ...
>
> We agree that distillation outcomes can depend on the distillation dataset. In this work, we analyse publicly released teacher-student checkpoints, so the dataset and the full distillation recipe are fixed and not under our control. As such, we have added a discussion to the limitations/future work section clarifying that certain findings should be interpreted as conditional on the data used during distillation.
>
> We have also added an Appendix (A) (referenced at start of Section 3), detailing training process and datasets for each student model used in our work.
>
> We also highlight that, as discussed in our response above to your first point, our circuit decompositions are intended as task- and model-specific case studies, while the higher-level compression/reliance/omission trends are the aspects we expect to generalise more broadly.
>
> ### Change: Add discussion of fixed distillation dataset limitations to Section 5.1
>
> ### Change: Add appendix detailing distillation training processes
>
> > For section 2.3, can the authors discuss the motivation for selecting these tasks? ...
>
> We aimed to find tasks suited for smaller models (GPT2 and BERT), and simple enough such that an in-depth analysis would be tractable. Additionally, the fact that these tasks have been studied in depth previously provides us with an established foundation, meaning our analysis is isolated in comparison of the teacher and student.
>
> We have since updated this list of tasks to include a more complex question answering task (simpleQA), which was run on the Llama model pair.
>
> > For section 4.1, can the authors define the proposed metric formally? ...
>
> ### Change: Update Section 4.1 with formalised definitions of each methodological step.
>
> ## References:
> [1] Lan, M., et al. 2024. Towards Interpretable Sequence Continuation: Analyzing Shared Circuits in Large Language Models. In Proceedings of the 2024 Conference on Empirical Methods in Natural Language Processing.
>
> [2] Conmy, A., et al. 2023. "Towards automated circuit discovery for mechanistic interpretability." Advances in Neural Information Processing Systems.

---

### Author Response · Authors · 2026-01-21
**New manuscript with changes made**

We would again like to thank all of the reviewers for their excellent feedback, which we believe has improved the strength and clarity of our work. We have now replied to all of the reviewers' concerns, and have submitted a revision of the paper with the following key changes made:

### Framing and clarity
- Clarified which findings are task/model-specific vs. expected to generalise (updates in Sec. 3.2, 4.2.2, 5)
- Discuss limitations around results being conditional on fixed distillation recipes / datasets (Sec. 5.1)
- Discuss limitations around using ablation robustness as a proxy for deployment robustness (Sec. 5.1)
- Formalised the alignment metric further (Sec. 4.1)
- Add practitioner guidance subsection (Sec. 4.3)
- Add computational costs for methodology (Sec. 3.1, 4.1)

### New tasks and experiments
- Added a more realistic downstream NLP task: SimpleQA on Llama-3.1-8B (teacher) and Minitron-4B (student), including setup and comparisons to prior trends (Sec. 2.3.3, 4.2.2, App. H.2)
- Add $T_n$ sweep (App. N)
- Add alignment metric design choice sensitivity analysis (App. M)

### Statistics, dataset sizes
- Increased dataset sizes, report 95% CIs (Sec. 3.2.3, Table 2, 4.2.2, Table 3)

### Future work
- Discuss work using interpretability agents to automate component role attribution (Sec. 5.1)
- Discuss work using alignment metric as a distillation loss term (Sec. 5.1)
- Motivate future work to further develop theoretical understanding of relationships between model capacity, robustness, and component redundancy (Sec. 5.1)

---

### Decision · Action_Editor_qSQB · 2026-03-03

**Recommendation:** Accept as is

**Audience:**

Yes

**Audience Explanation:**

Distillation is a key technique for training LLMs more efficiently. This work provides researchers on mechanistic interpretability, model compression, and robustness a new tool box. It allows the practitioners to understand when the student model collapses complex teacher circuits into a few brittle components. These techniques could be crucial for understanding the behavior of the student model and its generalization.

**Claims And Evidence:**

Yes

**Claims Explanation:**

The reviewers all reached a final agreement that the claims are supported. Reviewer yYWH, who was initially the most critical of the paper's generalization, explicitly stated that the additional studies on more complex tasks strengthen the claim. The new ablations and alignment scores provide a clearer view of the internal shifts caused by distillation.